# Neutralizing antibody correlate of protection against severe-critical COVID-19 in the ENSEMBLE single-dose Ad26.COV2.S vaccine efficacy trial

Assessment of immune correlates of severe COVID-19 has been hampered by the low numbers of severe cases in COVID-19 vaccine efficacy (VE) trials. We assess neutralizing and binding antibody levels at 4 weeks post-Ad26.COV2.S vaccination as correlates of risk and of protection against severe-critical COVID-19 through 220 days post-vaccination in the ENSEMBLE trial (NCT04505722), constituting ~4.5 months longer follow-up than our previous correlates analysis and enabling inclusion of 42 severe-critical vaccine-breakthrough cases. Neutralizing antibody titer is a strong inverse correlate of severe-critical COVID-19, with estimated hazard ratio (HR) per 10-fold increase 0.35 (95% CI: 0.13, 0.90). In a multivariable model, HRs are 0.31 (0.11, 0.89) for neutralizing antibody titer and 1.22 (0.49, 3.02) for anti-Spike binding antibody concentration. VE against severe-critical COVID-19 rises with neutralizing antibody titer: 63.1% (95% CI: 40.0%, 77.3%) at unquantifiable [<4.8975 International Units (IU)50/ml], 85.2% (47.2%, 95.3%) at just-quantifiable (5.2 IU50/ml), and 95.1% (81.1%, 96.9%) at 90th percentile (30.2 IU50/ml). At the same titers, VE against moderate COVID-19 is 32.5% (11.8%, 48.4%), 33.9% (19.1%, 59.3%), and 60.7% (40.4%, 76.4%). Protection against moderate vs. severe disease may require higher antibody levels, and very low antibody levels and/or other immune responses may associate with protection against severe disease.

Serum anti-SARS-CoV-2 neutralizing antibody (nAb) titer and serum anti-Spike binding antibody (bAb) concentration are supported as correlates of protection (CoPs)[1,2] against symptomatic SARS-CoV-2 infection[3]. However, the small numbers of severe COVID-19 cases in phase 3 COVID-19 vaccine efficacy (VE) trials have hindered characterization of CoPs against severe COVID-19, the most important outcome to prevent.

The ENSEMBLE trial was a randomized, placebo-controlled phase 3 trial of single-dose Ad26.COV2.S vaccine. A total of 44,325 participants were randomized 1:1 to receive Ad26.COV2.S or placebo on Day 1 (D1), with serum samples taken on D1 and D29 for antibody measurement (Supplementary Fig. 1). Results of the primary[4] and final[5] safety and efficacy analyses have been published. We previously showed that D29 50% inhibitory dilution neutralizing antibody titer (nAb-ID50), anti-Spike bAb concentration (Spike IgG), and anti-receptor binding domain bAb concentration (RBD IgG) were inverse correlates of risk (CoRs) of moderate to severe-critical COVID-19 through 83 days post-vaccination[6]. Correlate of protection (CoP) analyses provided strongest evidence for nAb-ID50 as a CoP[6].

Here we applied an identical approach using final data from the double-blind phase to assess the same antibody markers as CoRs and CoPs against severe-critical COVID-19 starting 7 days post-D29 through

e-mail: pgilbert@fredhutch.org

220 days post-vaccination, during which overall VE against severe-critical COVID-19 was 73.1% [95% confidence interval (CI) 58.7%, 84.1%]. We also assessed the same markers as correlates of moderate COVID-19 and of the primary endpoint in Sadoff et al.[5], moderate to severe-critical COVID-19, through 220 days, whereas all previous correlates analyses restricted to 83 days follow-up[6]. Overall VE against the moderate endpoint and against the primary endpoint starting 7 days post-D29 was 41.3% (28.6%, 51.3%) and 48.6% (38.6%, 57.0%), respectively. We focus on results for D29 nAb-ID50, and summarize results for D29 bAbs in the main text, with details in Supplementary Information. We repeated all analyses restricting to Latin America, South Africa, and the United States, except severe-critical COVID-19 could not be studied for the latter two regions due to too few events (Supplementary Table 1).

## Results

The correlates analyses used the final analysis database[5], with data cut-off July 9, 2021. The moderate, severe-critical, and moderate to severe-critical COVID-19 endpoints were defined as in Sadoff et al.[5], with minor differences as described in Methods. Correlates analyses were performed in per-protocol baseline SARS-CoV-2 seronegative participants, excluding participants with evidence of SARS-CoV-2 infection up to 6 days post-D29. Cases were participants with the relevant disease endpoint (onset both ≥ 28 days post-vaccination and ≥7 days post-D29) through to the cut-off date. Non-case vaccine recipients were sampled into the immunogenicity subcohort with no evidence of SARS-CoV-2 infection to the end of the correlates study period:

220 days post D1 (all regions, Latin America) or 140 days post D1 (South Africa, United States) but not later than the cut-off date.

Using a case-cohort design, participants were randomly sampled into an immunogenicity subcohort for D1 and D29 antibody measurements [see the Statistical Analysis Plan (SAP) for the previous ENSEMBLE correlates analyses[6]]. D1 and D29 antibodies were also measured from all moderate to severe-critical COVID-19 vaccine breakthrough cases (Supplementary Fig. 1). Supplementary Table 1 lists numbers of participants included in analyses; Supplementary Fig. 2 shows the study flowchart. Supplementary Table 2 provides demographic and clinical information of subcohort members (839 vaccine and 91 placebo recipients), and Supplementary Tables 3–5 provide region-specific breakdowns.

The SARS-CoV-2 variants causing the severe-critical cases varied over time and by region (Fig. 1, Supplementary Fig. 3). In Latin America, the most prevalent variants were Reference, Gamma, and Mu, causing 7, 9, and 4 of 23 cases in the vaccine arm and 29, 24, and 18 of 89 cases in the placebo arm, respectively. [As in Sadoff et al.[5], "Reference" refers to the index strain (GenBank accession number: MN908947.3) harboring the D614G point mutation.] Most severe-critical cases in the United States were Reference (2 of 4 cases in the vaccine arm, 16 of 20 cases in the placebo arm), and all in South Africa were Beta (14 placebo, 2 vaccine). Half (21) of the 42 severe-critical vaccine breakthrough cases had between 10 and 12 symptoms (Supplementary Tables 6, 7).

The proportion of vaccine recipients with quantifiable D29 nAb-ID50 titer [lower limit of quantitation = 4.8975 International Units (IU)/

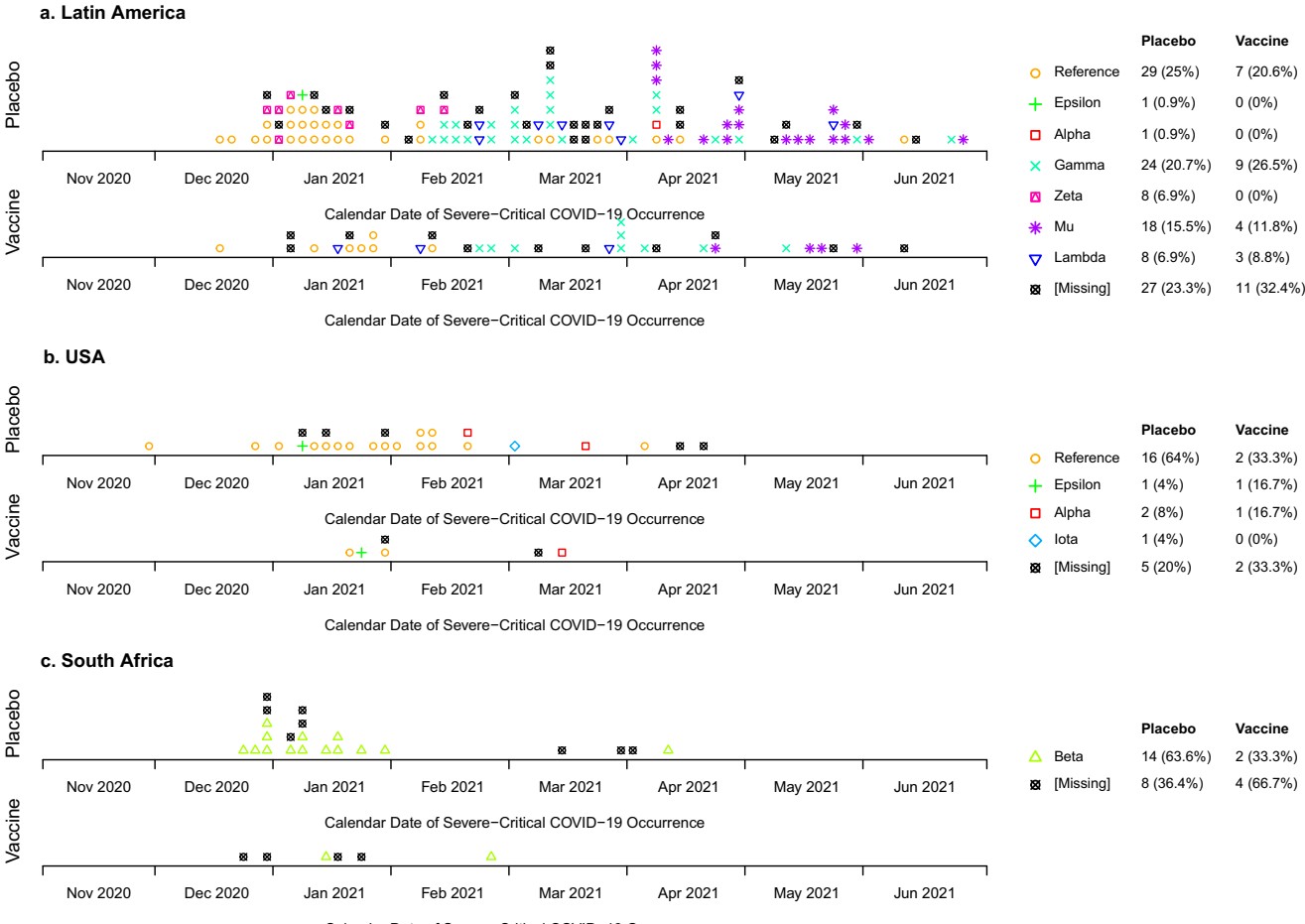

**Fig. 1 | SARS-CoV-2 variants causing the severe-critical COVID-19 endpoints.** Variants are shown by calendar date of severe-critical COVID-19 occurrence and are broken out by geographic region (**a**, Latin America; **b**, USA; **c**, South Africa) and treatment assignment. Endpoint counts do not require having D1 and D29 antibody marker data (see the flowchart provided as Supplementary Fig. 2). As in Sadoff et al.[5], "Reference" refers to the index strain (GenBank accession number: MN908947.3) harboring the D614G point mutation.

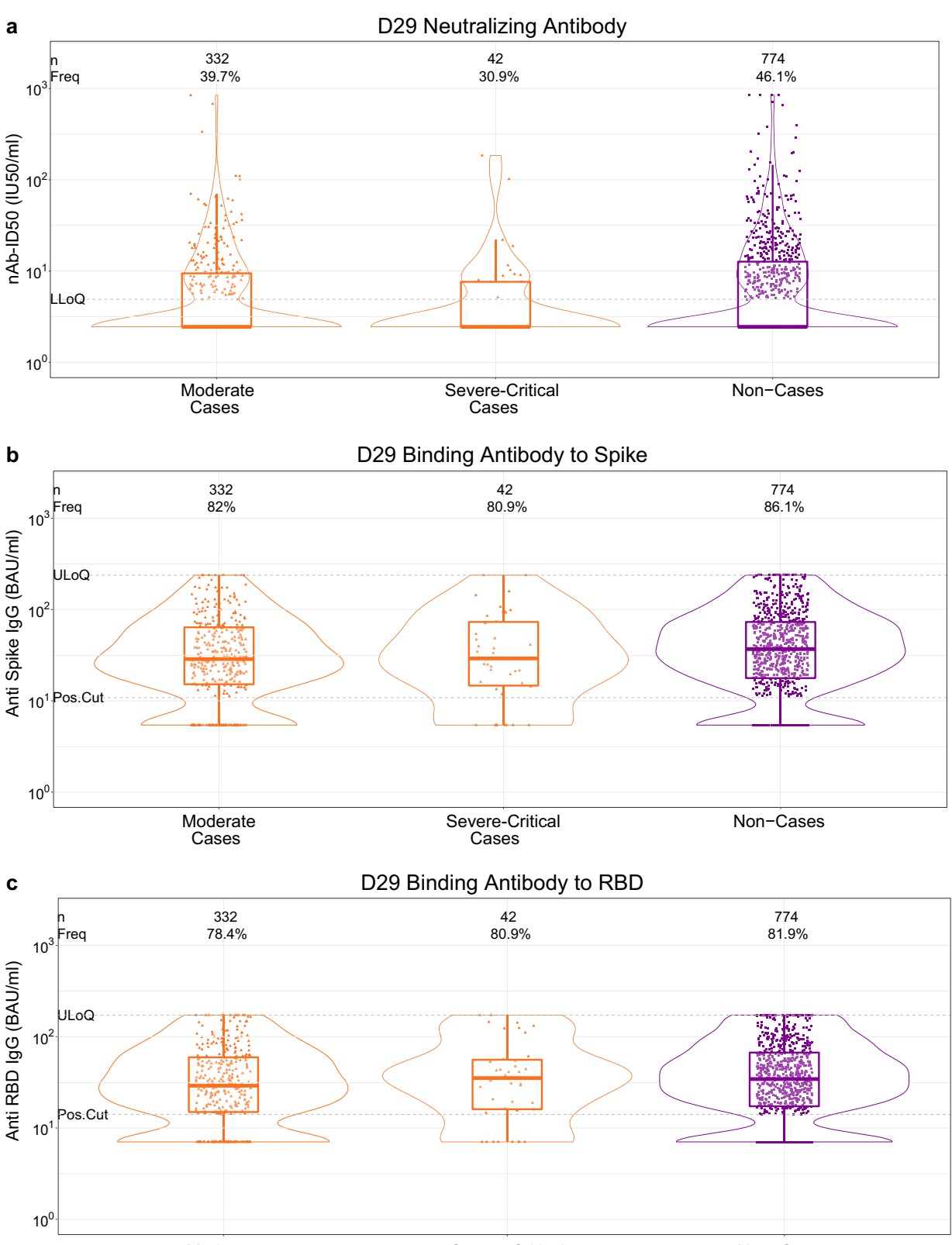

ml (IU50/ml)] was lowest in severe-critical cases (30.9%), intermediate in moderate cases (39.7%), and highest in non-cases (46.1%) (Fig. 2a). Geometric mean nAb-ID50 titers were 4.28 IU50/ml (95% CI: 3.15, 5.82), 5.02 IU50/ml (4.48, 5.63), and 6.06 IU50/ml (5.50, 6.67) in severe-critical cases, moderate cases, and non-cases, yielding a severe-critical case:non-case ratio of 0.71 (0.51, 0.98) and a moderate case:non-case

ratio of 0.83 (0.71, 0.96) (Table 1). D29 bAb response frequencies and levels were also lower in cases than in non-cases for both endpoints (Fig. 2b, c; Table 1).

In vaccine recipients, D29 nAb-ID50 and bAb levels correlated inversely with severe-critical COVID-19 risk and with moderate COVID-19 risk. The hazard ratio (HR) of severe-critical COVID-19 for the High

**Fig. 2 | D29 antibody marker level by COVID-19 outcome status (moderate COVID-19 case, severe-critical COVID-19 case, or non-case). a** 50% inhibitory dilution neutralizing antibody (nAb-ID50) titer, (**b**) anti-Spike IgG concentration, and (**c**) anti-RBD IgG concentration. Data points are from baseline SARS-CoV-2 seronegative per-protocol vaccine recipients. Violin plots contain interior box plots with upper and lower horizontal edges the 25th and 75th percentiles of antibody level and middle line the 50th percentile, and vertical bars the distance from the 25th (or 75th) percentile of antibody level and the minimum (or maximum) antibody level within the 25th (or 75th) percentile of antibody level minus (or plus) 1.5 times the interquartile range. Each side shows a rotated probability density (estimated by a kernel density estimator with a default Gaussian kernel) of the data. Positive response frequencies (Freq.) computed with inverse probability of sampling weighting. Positive response definitions: Spike IgG, IgG>10.8424 BAU/ml; RBD IgG, IgG>14.0858 BAU/ml. ULoQ: Spike IgG, 238.1165 BAU/ml; RBD IgG, 172.5755 BAU/ml. Positive response for nAb-ID50: D1 nAb-ID50 titer <LLOQ (LLOQ = 4.8975 IU50/ml) with detectable D29 nAb-ID50 ( ≥ LLOQ), or D1 nAb-ID50 > LLOQ with at least a fourfold increase in D29 nAb-ID50. ULoQ: ID50, 844.7208 IU50/ml. Moderate cases are baseline SARS-CoV-2 seronegative per-protocol vaccine recipients with the moderate COVID-19 endpoint (moderate COVID-19 with onset both ≥ 7 days post D29 and ≥28 days post-vaccination) up to 181 days post-D29 but not past data cut (July 9, 2021). Severe-critical cases are baseline SARS-CoV-2 seronegative per-protocol vaccine recipients with the severe-critical COVID-19 endpoint (severe-critical COVID-19 with onset both ≥ 7 days post-D29 and ≥28 days post-vaccination) up to 170 days post-D29 but not past data cut (July 9, 2021). Non-cases are baseline seronegative per-protocol vaccine recipients sampled into the immunogenicity subcohort with no evidence of SARS-CoV-2 infection up to the end of the correlates study period, which is up to 181 days post-D29 but not past data cut (July 9, 2021). BAU binding antibody units, IU international units, LLoQ lower limit of quantitation, Pos.Cut positivity cut-off, ULoQ upper limit of quantitation. Source data are provided as a Source Data file.

vs. Low nAb-ID50 tertiles was 0.21 (95% CI: 0.07, 0.67), with family-wise error rate (FWER) multiplicity-adjusted p value of 0.087 for a different hazard rate across the three tertile subgroups (Fig. 3e). For moderate COVID-19, the High vs. Low HR was 0.43 (0.25, 0.75), with FWER p = 0.052. Inverse correlations, less strong compared to those seen for nAb-ID50, were observed for the bAb markers with both COVID-19 endpoints (Fig. 3).

For the D29 quantitative markers, inverse correlations with endpoints were also observed, again stronger for severe-critical COVID-19, with HR per 10-fold increase in nAb-ID50 titer 0.35 (0.13, 0.90; FWER p = 0.098) compared to 0.53 (0.34, 0.82; FWER p = 0.031) for moderate COVID-19 (Table 2). Inverse correlations, less strong, were observed for both bAb markers with both COVID-19 endpoints (Table 2).

Cumulative incidence of severe-critical COVID-19 through 170 days post-D29 decreased across the analyzed ranges of vaccine recipient subgroups defined by D29 antibody levels at a specific value. For nAb-ID50, the cumulative incidence of severe-critical COVID-19 was estimated by a nonparametric method over values ranging from unquantifiable titer to the 90th percentile (30.2 IU50/ml). Estimated cumulative incidence was 0.70% (0.45%, 1.05%) at unquantifiable titer, 0.28% (0.07%, 0.90%) at just-quantifiable titer of 5.2 IU50/ml, and 0.09% (0.06%, 0.26%) at the 90th percentile titer 30.2 IU50/ml (blue curve, Supplementary Fig. 17c). Cumulative incidence of moderate COVID-19 also decreased: 5.2% (4.1%, 6.6%), 5.1% (3.2%, 5.9%), and 3.0% (1.8%, 4.4%) at the same values of unquantifiable titer, 5.2 IU50/ml, and 30.2 IU50/ml, respectively (blue curve, Supplementary Fig. 18c). A similar decrease in cumulative incidence with increasing concentration of each bAb marker was observed (Supplementary Figs. 17, 18).

We also estimated by nonparametric regression the cumulative incidence of severe-critical COVID-19 through 170 days post-D29 across ranges of vaccine recipient subgroups defined by D29 antibody levels exceeding a specific value. Cumulative incidence of severe-critical COVID-19 was 0.26% (0.18%, 0.34%) for all vaccine recipients; at the threshold 5.2 IU50/ml just above the nAb-ID50 assay's lower limit of quantitation (LLOQ, 4.8975 IU50/ml), estimated cumulative incidence decreased to 0.18% (0.074%, 0.29%) (Supplementary Fig. 25c). No further decrease in estimated cumulative incidence with increasing threshold was seen, even at nAb-ID50 thresholds much higher than the LLOQ. For moderate COVID-19, there was a smaller decrease in risk for all vaccine recipients vs. those with nAb-ID50 titer exceeding any threshold above the LLOQ (Supplementary Fig. 26c). Supplementary Figs. 25, 26 also show results for the bAb markers.

With a proportional hazards model including both D29 nAb-ID50 and D29 Spike IgG, the HR of severe-critical COVID-19 per 10-fold increase was 0.31 (0.11, 0.89; p = 0.029) for nAb-ID50 and 1.22 (0.49, 3.02; p = 0.67) for Spike IgG (Table 3), supporting nAb-ID50 as the independent correlate. A similar result was seen for moderate COVID-19: HR 0.55 (0.33, 0.92; p = 0.023) and 0.92 (0.57, 1.50; p = 0.74) per 10-fold increase for nAb-ID50 and Spike IgG, respectively (Table 3).

VE against severe-critical COVID-19 increased with D29 antibody level. For nAb-ID50, estimated VE at unquantifiable titer, just-quantifiable titer of 5.2 IU50/ml, and 90th percentile titer of 30.2 IU50/ml was 63.1% (40.0%, 77.3%), 85.2% (47.2%, 95.3%), and 95.1% (81.1%, 96.9%), respectively (Fig. 4a). In comparison, estimated VE against moderate COVID-19 at the same values of unquantifiable titer, 5.2 IU50/ml, and 30.2 IU50/ml was 32.5% (11.8%, 48.4%), 33.9% (19.1%, 59.3%), and 60.7% (40.4%, 76.4%), respectively (Fig. 4b). For Spike IgG, estimated VE against severe-critical COVID-19 at negative response, just-positive concentration of 11.1 BAU/ml, and 90th percentile concentration of 125 BAU/ml was 65.4% (25.6%, 83.9%), 69.8% (41.8%, 84.6%), and 82.0% (74.4%, 92.5%) (Fig. 4c). In comparison, estimated VE against moderate COVID-19 at the same values of negative response, 11.1 BAU/ml, and 125 BAU/ml was 14.8% (−36.2%, 46.7%), 32.7% (−13.2%, 53.2%), and 59.2% (53.4%, 64.1%), respectively (Fig. 4d).

Mediation analysis of the D29 markers showed that an estimated 28.6% (8.5%, 48.7%) of VE against severe-critical COVID-19 was mediated by nAb-ID50 titer (Table 4), with a similar proportion, 24.3% (−21.4%, 70.0%), mediated by Spike IgG concentration. In comparison, the estimated proportions of VE against moderate COVID-19 mediated by nAb-ID50 titer and Spike IgG concentration were 50.5% (0.8%, 100%) and 103% (−2.2%, 208%), with notably wide confidence intervals (Table 4).

Table 5 scorecards D29 antibody marker correlate performance[7], using three categories of correlate-quality criteria: (1) CoR, (2) CoP – VE modification, and (3) CoP – VE mediation (see "Methods"). (See Gilbert et al.[8] for a recent summary of four statistical frameworks for assessing immune CoPs, including the VE modification and VE mediation frameworks). We focused on three comparisons. In (A), each D29 marker was compared as a correlate of severe-critical vs. moderate COVID-19. In this comparison, Spike IgG ranked slightly better as a CoR of severe-critical COVID-19 than of moderate COVID-19, was an equally good VE Modification CoP against severe-critical COVID-19 and against moderate COVID-19, and was a better VE Mediation CoP against moderate COVID-19. RBD IgG ranked better as a CoR and as a CoP against moderate COVID-19, whereas nAb-ID50 ranked better as a CoR and a VE Modification CoP against severe-critical COVID-19 but better as a VE Mediation CoP against moderate COVID-19. In (B), the three D29 markers were compared as a correlate of moderate to severe-critical COVID-19. nAb-ID50 ranked as the best CoR and as the best VE Modification CoP, while Spike IgG was the best VE Mediation CoP. Comparison (C) repeated (B) for severe-critical COVID-19. nAb-ID50 ranked as the best CoR and CoP.

We also applied a third statistical framework for assessing CoPs, stochastic interventional VE (SVE)[9], to assess the D29 markers as CoPs against moderate to severe-critical COVID-19. In this framework, VE is estimated under hypothetical immune marker shifts applied to all individual vaccine recipients, relative to their observed immune marker levels. For D29 nAb-ID50, estimated VE generally increased with successive shifts in titer: At no D29 nAb-ID50 shift, estimated SVE was

**Table 1 | D29 antibody marker response frequencies and geometric means by COVID-19 outcome status, for all geographic regions pooled as well as separately by geographic region**

| D29 Marker | Severe-Critical COVID-19 Cases | | | Moderate COVID-19 Cases | | | Moderate to Severe-Critical COVID-19 Cases | | | Non-cases | | | Comparison: Severe-Critical COVID-19 Cases to Non-cases | | Comparison: Moderate COVID-19 Cases to Non-cases | |
|---|---|---|---|---|---|---|---|---|---|---|---|---|---|---|---|---|
| | N | Pos Resp Freq | GM (95% CI) | N | Pos Resp Freq | GM (95% CI) | N | Pos Resp Freq | GM (95% CI) | N | Pos Resp Freq | GM (95% CI) | Resp Freq Diff (Cases – Non-cases) | Ratio of GM (Cases/Non-cases) | Resp Freq Diff (Cases – Non-cases) | Ratio of GM (Cases/Non-cases) |
| **All geographic regions pooled** | | | | | | | | | | | | | | | | |
| nAb-ID50 (IU50/ml) | 42 | 30.9% (18.5, 46.8%) | 4.28 (3.15, 5.82) | 332 | 39.7% (34.5, 45.1%) | 5.02 (4.48, 5.63) | 373 | 38.5% (33.7, 43.6%) | 4.92 (4.42, 5.48) | 774 | 46.1% (41.9, 50.4%) | 6.06 (5.50, 6.67) | -15.2% (-28.3, 1.3%) | 0.71 (0.51, 0.98) | -6.5% (-13.2, 0.4%) | 0.83 (0.71, 0.96) |
| Spike IgG (BAU/ml) | 42 | 80.9% (65.6, 90.4%) | 29.35 (20.99, 41.05) | 332 | 82.0% (77.4, 85.7%) | 29.04 (25.92, 32.53) | 373 | 81.8% (77.5, 85.4%) | 28.98 (26.02, 32.28) | 774 | 86.1% (83.0, 88.8%) | 35.24 (32.23, 38.54) | -5.2% (-20.7, 4.8%) | 0.83 (0.59, 1.18) | -4.2% (-9.4, 0.7%) | 0.82 (0.71, 0.95) |
| RBD IgG (BAU/ml) | 42 | 80.9% (65.6, 90.4%) | 32.50 (24.00, 44.02) | 332 | 78.4% (73.6, 82.5%) | 28.65 (25.81, 31.80) | 373 | 78.6% (74.1, 82.5%) | 28.96 (26.23, 31.96) | 774 | 81.9% (78.4, 85.0%) | 33.49 (30.86, 36.34) | -1% (-16.6, 9.1%) | 0.97 (0.71, 1.33) | -3.5% (-9.2, 1.9%) | 0.86 (0.75, 0.98) |
| **Latin America** | | | | | | | | | | | | | | | | |
| nAb-ID50 (IU50/ml) | 31 | 25.7% (13.0, 44.5%) | 3.77 (2.74, 5.20) | 258 | 40.7% (34.8, 46.8%) | 5.28 (4.60, 6.05) | 288 | 38.9% (33.4, 44.7%) | 5.08 (4.47, 5.77) | 197 | 52.7% (44.8, 60.5%) | 7.03 (5.85, 8.44) | -27% (-41.9, -6.7%) | 0.54 (0.37, 0.78) | -12% (-21.8, -2%) | 0.75 (0.60, 0.94) |
| Spike IgG (BAU/ml) | 31 | 80.6% (62.1, 91.3%) | 29.65 (20.28, 43.35) | 258 | 81.8% (76.5, 86.0%) | 28.89 (25.34, 32.94) | 288 | 81.6% (76.6, 85.7%) | 28.85 (25.48, 32.66) | 197 | 86.4% (80.7, 90.7%) | 34.90 (29.87, 40.78) | -5.8% (-24.8, 6.3%) | 0.85 (0.56, 1.28) | -4.7% (-11.4, 2.5%) | 0.83 (0.68, 1.01) |
| RBD IgG (BAU/ml) | 31 | 80.6% (62.1, 91.3%) | 32.21 (22.90, 45.32) | 258 | 78.6% (73.2, 83.2%) | 28.36 (25.21, 31.91) | 288 | 78.8% (73.7, 83.1%) | 28.62 (25.60, 31.99) | 197 | 84.0% (77.7, 88.8%) | 33.80 (29.31, 38.96) | -3.4% (-22.5, 9.1%) | 0.95 (0.66, 1.38) | -5.3% (-12.6, 2.5%) | 0.84 (0.70, 1.01) |
| **South Africa** | | | | | | | | | | | | | | | | |
| nAb-ID50 (IU50/ml) | | | | | | | 18 | 42.8% (21.3, 67.4%) | 5.64 (3.36, 9.46) | 181 | 43.2% (35.0, 51.8%) | 6.18 (4.99, 7.65) | | | | |
| Spike IgG (BAU/ml) | | | | | | | 18 | 84.0% (57.6, 95.3%) | 35.44 (21.34, 58.87) | 181 | 88.5% (82.4, 92.6%) | 39.40 (32.89, 47.20) | | | | |
| RBD IgG (BAU/ml) | | | | | | | 18 | 74.9% (46.4, 91.1%) | 36.29 (21.07, 62.52) | 181 | 84.7% (77.8, 89.7%) | 36.29 (30.82, 42.72) | | | | |
| **United States** | | | | | | | | | | | | | | | | |
| nAb-ID50 (IU50/ml) | | | | | | | 67 | 35.9% (25.1, 48.2%) | 4.13 (3.45, 4.94) | 396 | 41.3% (35.8, 46.9%) | 5.31 (4.72, 5.97) | | | | |
| Spike IgG (BAU/ml) | | | | | | | 67 | 82.1% (70.8, 89.7%) | 27.87 (22.01, 35.29) | 396 | 85.3% (80.7, 89.0%) | 34.61 (30.56, 39.19) | | | | |
| RBD IgG (BAU/ml) | | | | | | | 67 | 79.1% (67.5, 87.4%) | 28.51 (22.80, 35.65) | 396 | 79.5% (74.4, 83.9%) | 32.61 (29.10, 36.55) | | | | |

Analysis based on baseline SARS-CoV-2 seronegative per-protocol vaccine recipients. Median (interquartile range) days from vaccination to D29 was 29 (2) for all regions pooled, 29 (2) for the United States, 29 (3) for Latin America, and 29 (2) for South Africa. *BAU* antibody binding units, *CI* confidence interval, *GM* geometric mean, *IU* international units, *nAb-ID50* 50% inhibitory dilution neutralizing antibody, *Pos Resp Freq* positive response frequency, *RBD* receptor binding domain, *Resp Freq Diff* response frequency difference.

For the binding antibody assays, positive response was defined by having an IgG concentration above the specified positivity cut-off, with a separate cut-off for each antigen (10.8424 BAU/ml for Spike and 14.0858 BAU/ml for RBD; Table S10). Positive response for nAb-ID50 was defined by a D1 nAb-ID50 titer <LLOQ (LLOQ = 4.8975 IU50/ml) with quantifiable D29 nAb-ID50 (≥ LLOQ), or by D1 nAb-ID50 > LLOQ with at least a fourfold increase in D29 nAb-ID50.

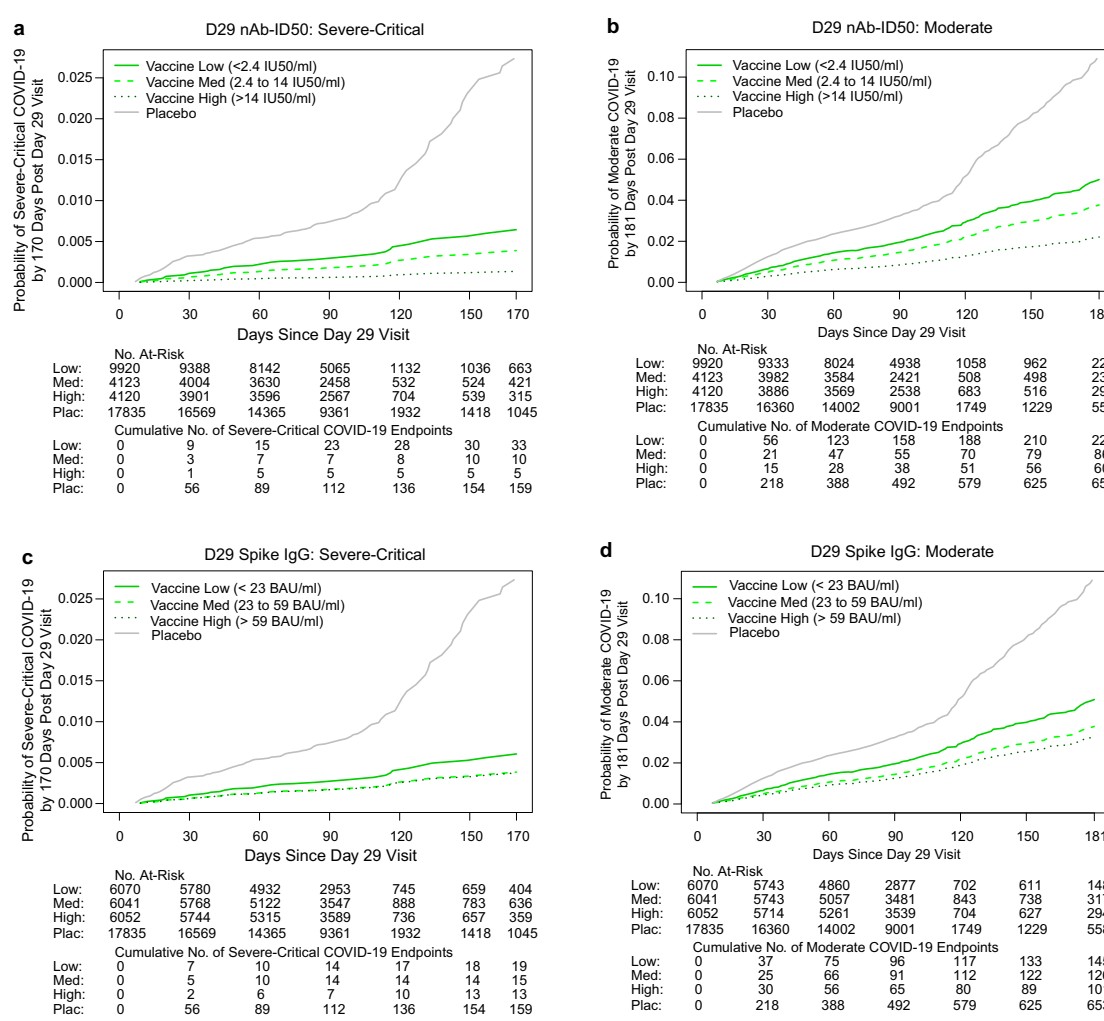

No. At-Risk = estimated number in the population for analysis: baseline seronegative per-protocol vaccine recipients not experiencing the (a, c) severe-critical or (b, c) moderate COVID-19 endpoint or with any evidence of infection through D29.
Cumulative No. of COVID-19 Endpoints = estimated cumulative number of this cohort with a (A, C) severe-critical or (B, D) moderate COVID-19 endpoint.

| D29 Immunologic Marker | COVID-19 Endpoint | Tertile | No. cases / No. at-risk | Attack rate | Hazard Ratio Pt. Est. | Hazard Ratio 95% CI | P-value (2-sided) | Overall P-value | FDR-adjusted P-value | FWER-adjusted P-value |
|---|---|---|---|---|---|---|---|---|---|---|
| nAb-ID50 (IU50/ml) | Severe-Critical | Low | 33/9,920 | 0.0033 | 1 | N/A | N/A | 0.025 | 0.106 | 0.087 |
| | | Med | 10/4,123 | 0.0024 | 0.60 | (0.26, 1.42) | 0.247 | | | |
| | | High | 5/4,120 | 0.0012 | 0.21 | (0.07, 0.67) | 0.008 | | | |
| nAb-ID50 (IU50/ml) | Moderate | Low | 225/9,920 | 0.0227 | 1 | N/A | N/A | 0.012 | 0.054 | 0.052 |
| | | Med | 88/4,123 | 0.0213 | 0.75 | (0.46, 1.21) | 0.231 | | | |
| | | High | 61/4,120 | 0.0148 | 0.43 | (0.25, 0.75) | 0.003 | | | |
| Spike IgG (BAU/ml) | Severe-Critical | Low | 19/6,070 | 0.0031 | 1 | N/A | N/A | 0.411 | 0.633 | 0.723 |
| | | Med | 15/6,041 | 0.0025 | 0.63 | (0.28, 1.43) | 0.271 | | | |
| | | High | 13/6,052 | 0.0021 | 0.62 | (0.27, 1.42) | 0.254 | | | |
| Spike IgG (BAU/ml) | Moderate | Low | 145/6,070 | 0.0239 | 1 | N/A | N/A | 0.142 | 0.169 | 0.220 |
| | | Med | 127/6,041 | 0.0210 | 0.73 | (0.46, 1.16) | 0.188 | | | |
| | | High | 102/6,052 | 0.0169 | 0.64 | (0.40, 1.01) | 0.057 | | | |
| RBD IgG (BAU/ml) | Severe-Critical | Low | 17/6,062 | 0.0028 | 1 | N/A | N/A | 0.632 | 0.633 | 0.793 |
| | | Med | 16/6,066 | 0.0026 | 0.78 | (0.35, 1.73) | 0.534 | | | |
| | | High | 15/6,035 | 0.0025 | 0.67 | (0.28, 1.57) | 0.352 | | | |
| RBD IgG (BAU/ml) | Moderate | Low | 144/6,062 | 0.0238 | 1 | N/A | N/A | 0.085 | 0.122 | 0.173 |
| | | Med | 118/6,066 | 0.0195 | 0.67 | (0.42, 1.05) | 0.079 | | | |
| | | High | 112/6,035 | 0.0186 | 0.62 | (0.39, 0.99) | 0.047 | | | |
| Placebo | Severe-Critical | | 163/17,835 | 0.0091 | | | | | | |
| Placebo | Moderate | | 657/17,835 | 0.0368 | | | | | | |

47.7% (95% CI: 44.6%, 50.7%), and with 1.6-fold, 4-fold, and 10-fold shifts, estimated SVE was 57.3% (53.4%, 60.8%), 54.4% (47.9%, 60.1%), and 62.9% (54.2%, 69.9%), respectively (Supplementary Fig. 49c). The p-value for testing the hypothesis that VE changes as a function of shift in D29 nAb-ID50 titer (see Methods) was <0.001, providing further evidence in support of D29 nAb-ID50 as a CoP against moderate to severe-critical COVID-19. A similar result was seen for D29 RBD IgG, with estimated SVE increasing to 49.7% (46.0%, 53.1%), 58.5% (51.5%, 64.5%), and 69.7% (54.0%, 80.0%), respectively, at the same shifts of 1.6-fold, fourfold, and tenfold in IgG concentration (p = 0.007 for testing the hypothesis that VE changes as a function of shift in D29 RBD IgG concentration) (Supplementary Fig. 49b). For D29 Spike IgG, the

**Fig. 3 | Severe-critical COVID-19 risk and moderate COVID-19 risk by D29 antibody marker tertile.** Plots show covariate-adjusted cumulative incidence of (**a, c** severe-critical COVID-19 or **b, d**) moderate COVID-19 by Low, Medium, and High tertiles of D29 (**a, b**) 50% inhibitory dilution neutralizing antibody titer (nAb-ID50) or (**b, d**) anti-Spike IgG concentration in baseline SARS-CoV-2–seronegative per-protocol vaccine recipients. **e** Covariate-adjusted hazard ratios of severe-critical COVID-19 or of moderate COVID-19 across D29 antibody marker tertiles.

Endpoint counts for (**a–d**) calculated by inverse probability of sampling D29 marker weighting. The overall p value is from a two-sided generalized Wald test of whether the hazard rate of the designated COVID-19 endpoint differed across the Low, Medium, and High subgroups. Analyses adjusted for baseline behavioral risk score and geographic region. BAU binding antibody units, CI confidence interval, FDR false discovery rate, FWER family-wise error rate, IU international units, Pt. Est. point estimate. Source data are provided as a Source Data file.

## Table 2 | Covariate-adjusted hazard ratios of severe-critical COVID-19 or of moderate COVID-19 per tenfold increase or per standard deviation increase in each D29 antibody marker

| | **Severe-critical COVID-19** | | | | | | | |
|---|---|---|---|---|---|---|---|---|
| D29 Marker | No. cases/No. at-risk | HR per 10-fold increase | | p-value (2-sided) | FDR-adj p-value | FWER-adj p-value | HR per SD increase | |
| | | Pt. Est. | 95% CI | | | | Pt. Est. | 95% CI |
| nAb-ID50 (IU50/ml) | 46/18,163 | 0.35 | 0.13, 0.90 | 0.030 | 0.106 | 0.098 | 0.59 | 0.36, 0.95 |
| Spike IgG (BAU/ml) | 46/18,163 | 0.67 | 0.32, 1.39 | 0.285 | 0.619 | 0.567 | 0.83 | 0.59, 1.17 |
| RBD IgG (BAU/ml) | 46/18,163 | 0.79 | 0.33, 1.85 | 0.583 | 0.633 | 0.793 | 0.90 | 0.63, 1.30 |
| | **Moderate COVID-19** | | | | | | | |
| D29 Marker | No. cases/No. at-risk* | HR per 10-fold increase | | p-value (2-sided) | FDR-adj p-value** | FWER-adj p-value** | HR per SD increase | |
| | | Pt. Est. | 95% CI | | | | Pt. Est. | 95% CI |
| nAb-ID50 (IU50/ml) | 375/18,163 | 0.53 | 0.34, 0.82 | 0.005 | 0.052 | 0.031 | 0.73 | 0.58, 0.91 |
| Spike IgG (BAU/ml) | 375/18,163 | 0.67 | 0.45, 1.01 | 0.057 | 0.114 | 0.146 | 0.83 | 0.69, 1.01 |
| RBD IgG (BAU/ml) | 375/18,163 | 0.59 | 0.37, 0.93 | 0.025 | 0.073 | 0.076 | 0.80 | 0.65, 0.97 |

Analyses were based on baseline SARS-CoV-2 seronegative per-protocol vaccine recipients and adjusted for baseline behavioral risk score and geographic region.
*.No. at-risk = estimated number in the population for analysis, i.e. baseline negative per-protocol vaccine recipients not experiencing the designated COVID-19 endpoint or infected through 6 days post Day 29 visit; no. cases = number of this cohort with an observed designated COVID-19 endpoint (calculated via inverse probability of sampling Day 29 marker weighting).
**q-value and FWER (family-wide error rate) are computed over the set of p values both for quantitative markers and categorical markers using the Westfall and Young permutation method (10000 replicates).
p-values were obtained using a two-sided Wald test.
Cases were counted starting 7 days post Day 29.
BAU antibody binding units, CI confidence interval, FDR false discovery rate, FWER family-wise error rate, HR hazard ratio, IU international units, nAb-ID50 50% inhibitory dilution neutralizing antibody, Pt. Est. point estimate, RBD receptor binding domain, SD standard deviation.

## Table 3 | Covariate-adjusted hazard ratios, assessed using multivariable models, of severe-critical COVID-19 or of moderate COVID-19 per tenfold increase in each D29 antibody marker

| | **Severe-critical COVID-19** | | **Moderate COVID-19** | |
|---|---|---|---|---|
| | Hazard ratio (95% CI) | P value | Hazard ratio (95% CI) | P value |
| Risk score | 2.27 (1.28, 4.04) | 0.005 | 1.59 (1.16, 2.19) | 0.004 |
| Region: Latin America* | 1.56 (0.50, 4.82) | 0.441 | 1.82 (1.19, 2.79) | 0.006 |
| Region: South Africa* | 2.57 (0.77, 8.55) | 0.124 | 0.65 (0.36, 1.16) | 0.145 |
| nAb-ID50 (IU50/ml) | 0.31 (0.11, 0.89) | 0.029 | 0.55 (0.33, 0.92) | 0.023 |
| Spike IgG (BAU/ml) | 1.22 (0.49, 3.02) | 0.674 | 0.92 (0.57, 1.50) | 0.741 |

Analyses were based on baseline SARS-CoV-2 seronegative per-protocol vaccine recipients and adjusted for baseline behavioral risk score and geographic region.
*Reference region = United States.
Maximum failure event time 170 days (severe-critical COVID-19) or 181 days (moderate COVID-19) post D29. Cases were counted starting 7 days post D29.
P values are unadjusted and were obtained using a two-sided Wald test.
BAU antibody binding units, CI confidence interval, IU international units, nAb-ID50 50% inhibitory dilution neutralizing antibody.

increases in SVE were smaller with shifted IgG concentration and the p-value for testing the hypothesis that VE changes as a function of shift in D29 Spike IgG concentration was 0.12 (Supplementary Fig. 49a).

Region-specific correlates analyses of severe-critical COVID-19 could only be conducted for Latin America (31 severe-critical vaccine endpoints) due to the low numbers of severe-critical vaccine endpoints in South Africa (5) and the United States (6) (Supplementary Table 1). While the Latin America-specific results for assessing D29 markers as correlates were similar to those of the region-pooled analyses, the point estimates for the severe-critical endpoint tended to indicate stronger correlates and the p-values tended to indicate greater significance. For example, the HR in the Latin America cohort of severe-critical COVID-19 per 10-fold increase in nAb-ID50 was 0.20 (0.05, 0.73; FWER-adjusted p = 0.048) and for moderate COVID-19 it

was 0.53 (0.32, 0.90; FWER-adjusted p = 0.085) (Supplementary Table 12). Full results of the Latin America-specific analyses are reported in the Supplementary Text.

Given that this analysis assesses immune correlates through ~7 months post-vaccination, whereas our previous correlates analysis of ENSEMBLE[5] assessed through ~2.5 months post-vaccination, waning of antibody levels over time is important to consider. Using a measurement error statistical method, we performed an exposure-proximal correlates analysis for a hypothetical scenario where the antibody marker under study was repeatedly measured from serum samples collected on every day of follow-up, and the analysis assesses how the current value of this daily measured marker correlates with the hazard of COVID-19 (i.e., the probability of COVID-19 occurrence over the next day) (see Methods for details). From these current-

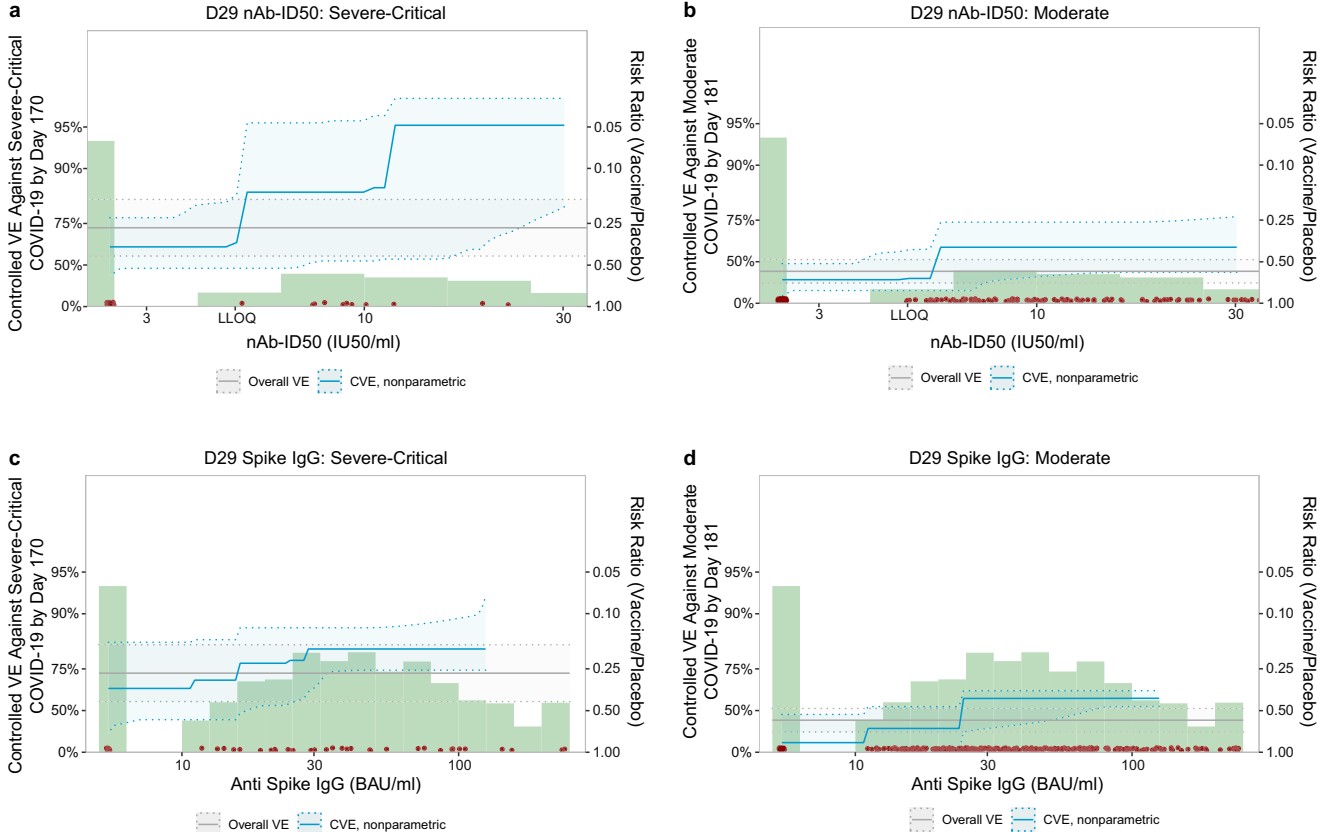

**Fig. 4 | Vaccine efficacy against severe-critical COVID-19 or against moderate COVID-19 by D29 antibody marker level.** Vaccine efficacy estimates against (**a**, **c**) severe-critical COVID-19 and against (**b**, **d**) moderate COVID-19 through 170 (severe-critical) or 181 (moderate) days post-D29 were obtained using a nonparametric implementation of the method of Gilbert et al.[30]. Each point on the curve represents the estimated controlled vaccine efficacy at the given D29 antibody marker level: (**a**, **b**) 50% inhibitory dilution neutralizing antibody (nAb-ID50) titer and (**c**, **d**) anti-Spike IgG binding antibody concentration. Dotted lines indicate bootstrap pointwise 95% CIs. The green histograms are frequency distributions of D29 marker level, with maroon dots representing marker levels of individual cases. Analyses adjusted for baseline behavioral risk score and geographic region. Curves are plotted over the nAb-ID50 titer range from unquantifiable to the 90th percentile (30.2 IU50/ml) and over the Spike IgG concentration range from negative response to the 90th percentile (125 BAU/ml). The horizontal gray line is the overall vaccine efficacy against (**a**, **c**) severe-critical COVID-19 or against (**b**, **d**) moderate COVID-19 through 170 (severe-critical) or 181 (moderate) days post-D29, with the dotted gray lines indicating the 95% CIs. BAU binding antibody units, CVE controlled vaccine efficacy, IU international units, LLOQ lower limit of quantitation, nAb-ID50 50% inhibitory dilution neutralizing antibody. nAb-ID50 LLOQ = 4.8975 IU50/ml; Spike IgG positivity cutoff = 10.8424 BAU/ml. Source data are provided as a Source Data file.

**Table 4 | Mediation effect estimates for D29 quantitative markers with 95% confidence intervals**

| VE against severe-critical COVID-19 | | | |
|---|---|---|---|
| | Non-Marker Mediated VE (95% CI) | Marker Mediated VE (95% CI) | Prop. Mediated (95% CI) |
| nAb-ID50 (IU50/ml) | 0.645 (0.311, 0.817) | 0.340 (0.166, 0.477) | 0.286 (0.085, 0.487) |
| Spike IgG (BAU/ml) | 0.667 (0.191, 0.863) | 0.297 (−0.349, 0.634) | 0.243 (−0.214, 0.700) |
| RBD IgG (BAU/ml) | 0.696 (0.257, 0.876) | 0.228 (−0.497, 0.602) | 0.179 (−0.282, 0.639) |
| **VE against moderate COVID-19** | | | |
| | Non-Marker Mediated VE (95% CI) | Marker Mediated VE (95% CI) | Prop. Mediated (95% CI) |
| nAb-ID50 (IU50/ml) | 0.196 (−0.048, 0.383) | 0.199 (0.017, 0.348) | 0.505 (0.008, 1.00) |
| Spike IgG (BAU/ml) | −0.013 (−0.605, 0.361) | 0.365 (0.031, 0.583) | 1.03 (−0.022, 2.08) |
| RBD IgG (BAU/ml) | 0.063 (−0.410, 0.377) | 0.313 (0.002, 0.527) | 0.852 (−0.051, 1.76) |

Non-marker mediated VE = VE comparing vaccine vs. placebo with antibody marker set to value if assigned placebo.

Marker-mediated VE = VE in vaccinated comparing observed antibody marker vs. hypothetical marker had the participant received placebo.

Prop. Mediated = fraction of total risk reduction from vaccine attributed to the antibody marker.

BAU, antibody binding units; IU, international units; nAb-ID50, 50% inhibitory dilution neutralizing antibody.

Overall VE (95% CI) against the severe-critical, moderate, and moderate to severe-critical endpoints starting 7 days post-D29 through 220 days post-vaccination was 73.1% (58.7%, 84.1%), 41.3% (28.6%, 51.3%), and 48.6% (38.6%, 57.0%), respectively.

Proportion mediated is not a true proportion in that it can take values outside of [0, 1]. Proportion mediated = 1 is equivalent to Non-marker-mediated VE = 0% and Proportion mediated = 0 is equivalent to Non-marker-mediated VE = Overall VE.

**Table 5 | Scorecard for ranking D29 antibody marker performance in each of three categories of immune correlate-quality criteria**

| | | | Category 1: correlate of risk (CoR) | | | | | Category 2: correlate of protection (CoP): VE modification | | | | | | | Category 3: CoP: VE mediation | | | | |
|---|---|---|---|---|---|---|---|---|---|---|---|---|---|---|---|---|---|---|---|
| | | | HR per SD (Cox, quant.) | | HR High vs. Low tertile (Cox) | | Cat. 1 Mean rank | Fold-increase in CVE† (Cox) | | Fold-increase in CVE† (NP) | | E-val. Marg. RR Pt. Est. | | Cat. 2 Mean rank | Proportion mediated | | | | Cat. 3 Mean rank |
| Comp. | D29 Marker | COVID-19 endpoint | Pt. Est. (95% CI) | Rank | Pt. Est. (95% CI) | Rank | | Fold-increase | Rank | Fold-increase | Rank | E-val. | Rank | | Pt. Est. (95% CI) | Rank | 95% LCL | Rank | |
| | Spike IgG (BAU/ml) | Sev-crit | 0.83 (0.59, 1.17) | 1 | 0.62 (0.27, 1.42) | 1 | 1* | 1.7 | 1 | 1.9 | 1 | 2.6 | 1 | 1.33* | 0.243 (−0.214, 0.700) | 2 | −0.214 | NA | 2 |
| A | Spike IgG (BAU/ml) | Mod | 0.83 (0.69, 1.01) | 1 | 0.64 (0.40, 1.01) | 2 | 1.5 | 1.7 | 1 | 2.1 | 1 | 2.5 | 2 | 1.33* | 1.03 (−0.022, 2.08) | 1 | −0.022 | NA | 1* |
| | RBD IgG (BAU/ml) | Sev-crit | 0.90 (0.63, 1.30) | 2 | 0.67 (0.28, 1.57) | 2 | 2 | 1.3 | 2 | 1.8 | 2 | 2.4 | 2 | 2 | 0.179 (−0.28, 0.639) | 2 | −0.282 | NA | 2 |
| A | RBD IgG (BAU/ml) | Mod | 0.80 (0.65, 0.97) | 1 | 0.62 (0.39, 0.99) | 1 | 1* | 1.9 | 1 | 1.9 | 1 | 2.6 | 1 | 1* | 0.852 (−0.051, 1.76) | 1 | −0.051 | NA | 1* |
| | nAb-ID50 (IU50/ml) | Sev-crit | 0.59 (0.36, 0.95) | 1 | 0.21 (0.07, 0.67) | 1 | 1* | 3.2 | 1 | 7.6 | 1 | 9.0 | 1 | 1* | 0.286 (0.085, 0.487) | 2 | 0.085 | NA | 2 |
| A | nAb-ID50 (IU50/ml) | Mod | 0.73 (0.58, 0.91) | 2 | 0.43 (0.25, 0.75) | 2 | 2 | 2.0 | 2 | 1.7 | 2 | 4.0 | 2 | 2 | 0.505 (0.008, 1.00) | 1 | 0.008 | NA | 1* |
| A | Spike IgG (BAU/ml) | Mod to sev-crit | 0.83 (0.69, 1.00) | 3 | 0.63 (0.40, 0.99) | 3 | 3 | 1.7 | 3 | 2.1 | 3 | 2.5 | 3 | 2.33 | 0.723 (0.051, 1.39) | 1 | 0.051 | 2 | 1.5* |
| B | RBD IgG (BAU/ml) | Mod to sev-crit | 0.80 (0.66, 0.98) | 2 | 0.62 (0.39, 0.98) | 2 | 2 | 1.8 | 2 | 1.9 | 2 | 2.6 | 2 | 2 | 0.603 (0.015, 1.19) | 2 | 0.015 | 3 | 2.5 |
| | nAb-ID50 (IU50/ml) | Mod to sev-crit | 0.71 (0.57, 0.88) | 1 | 0.40 (0.23, 0.69) | 1 | 1* | 2.1 | 1 | 1.9 | 1 | 4.3 | 1 | 1.33* | 0.415 (0.088, 0.742) | 3 | 0.088 | 1 | 2 |
| C | Spike IgG (BAU/ml) | Sev-crit | 0.83 (0.59, 1.17) | 2 | 0.62 (0.27, 1.42) | 2 | 2 | 1.7 | 2 | 1.9 | 2 | 2.6 | 2 | 2 | 0.243 (−0.214, 0.700) | 2 | −0.214 | 2 | 2 |
| | RBD IgG (BAU/ml) | Sev-crit | 0.90 (0.63, 1.30) | 3 | 0.67 (0.28, 1.57) | 3 | 3 | 1.3 | 3 | 1.8 | 3 | 2.4 | 3 | 3 | 0.179 (−0.282, 0.639) | 3 | −0.282 | 3 | 3 |
| | nAb ID50 (IU50/ml) | Sev-crit | 0.59 (0.36, 0.95) | 1 | 0.21 (0.07, 0.67) | 1 | 1* | 3.2 | 1 | 7.6 | 1 | 9.0 | 1 | 1* | 0.286 (0.085, 0.487) | 1 | 0.085 | 1 | 1* |

†Fold-increase calculated on the [1-VE(unquantifiable or negative)]/[1-VE(90th percentile)] scale.

*Antibody marker(s) with the best performance, within each comparison, within each category.

Bold font indicates the three categories of criteria and the mean D29 antibody marker rank within each category.

Baseline covariates adjusted for: Baseline risk score, geographic region (Latin America, South Africa, United States). Maximum failure event time 181 days (moderate COVID-19),170 days (severe-critical COVID-19) post D29. Cases were counted starting 7 days post D29. All serological assay readouts were expressed in WHO International Standard units (see "Methods"). The Proportion mediated is not a true proportion in that it can take values outside of the interval [0, 1]. Proportion mediated = 1 is equivalent to Non-marker-mediated VE = 0% and Proportion mediated = 0 is equivalent to Non-marker-mediated VE = Overall VE. BAU binding antibody units, Cat. category, CI confidence interval, Comp. comparison, E-val. E-value, HR hazard ratio, IU international units, LCL Lower confidence limit, Marg. marginalized, mod. moderate, NP nonparametric, nAb-ID50 50% inhibitory dilution neutralizing antibody titer, Pt. Est. point estimate, RR risk ratio, SD standard deviation, sev-crit severe-critical.

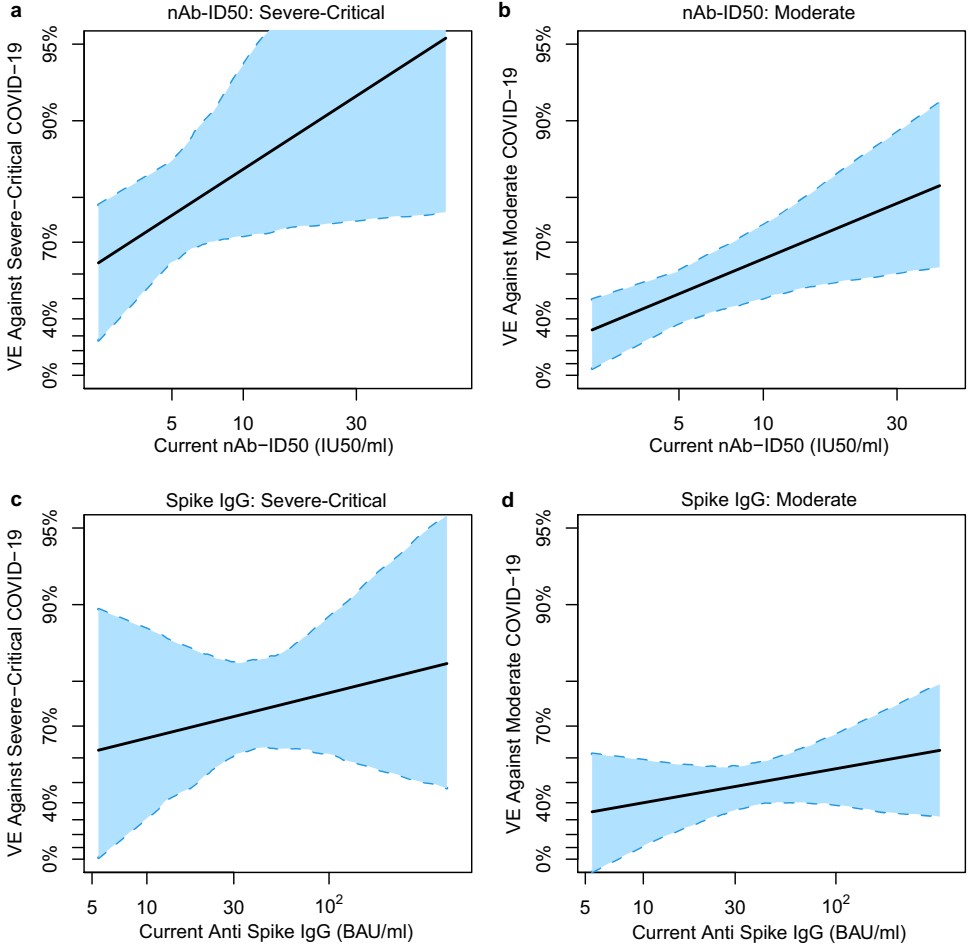

**Fig. 5 | Exposure-proximal vaccine efficacy against severe-critical COVID-19 or against moderate COVID-19 by current antibody marker level.** Analyses were performed in baseline SARS-CoV-2 seronegative per-protocol vaccine recipients. Exposure-proximal vaccine efficacy estimates against (**a, c**) severe-critical COVID-19 and against (**b, d**) moderate COVID-19 through 170 (severe-critical) or 181 (moderate) days post-D29 by current antibody marker level were obtained using the method of Huang and Follmann[44], with "current" referring to the true underlying antibody marker level not subject to technical measurement error, in a hypothetical scenario in which the value was available from serum samples collected every day over the follow-up period (see "Methods"). Each point on the curve represents the vaccine efficacy at the given current antibody marker level: (**a, b**) 50% inhibitory dilution neutralizing antibody (nAb-ID50) titer and (**c, d**) anti-Spike IgG binding antibody concentration. The dashed lines are bootstrap pointwise 95% CIs. Analyses adjusted for baseline behavioral risk score and geographic region. Curves are plotted over the range from negative binding antibody response (or unquantifiable neutralizing antibody titer) to the 97.5th percentile of each current antibody marker level: Spike IgG, negative response to 352 BAU/ml; RBD IgG, negative response to 486 BAU/ml; nAb-ID50, unquantifiable to 43.4 IU50/ml. Positivity cutoffs: 10.8424 BAU/ml for Spike and 14.0858 BAU/ml for RBD; nAb-ID50 LLOQ = 4.8975 IU50/ml. BAU binding antibody units, CI confidence interval, IU international units, LLOQ lower limit of quantitation, nAb-ID50 50% inhibitory dilution neutralizing antibody. Source data are provided as a Source Data file.

marker conditional hazard curves, we generated current-marker conditional VE curves (exposure-proximal VE) by dividing the conditional hazard curve by the hazard of COVID-19 for the whole placebo arm. Figure 5a shows that exposure-proximal VE against severe-critical COVID-19 rose as current nAb-ID50 titer increased across the range of analyzed values (unquantifiable titer up to the 97.5th percentile). Similar results were obtained for current Spike IgG concentration, albeit with a less steep increase and with a wider 95% CI at the left end of the curve (Fig. 5c). Similarly, exposure-proximal VE against moderate COVID-19 increased with current nAb-ID50 titer (Fig. 5b) as well as with current Spike IgG concentration (Fig. 5d). Latin America-specific exposure-proximal VE curves against severe-critical COVID-19 are shown in Supplementary Fig. 54 and against moderate COVID-19 in Supplementary Fig. 55; these results, which were similar to those in the pooled analysis, are discussed in the Supplementary Text.

## Discussion
The substantial estimated VE against severe-critical COVID-19 (63.1% at unquantifiable nAb-ID50 titer and 65.4% at negative Spike IgG

response) for vaccine recipients with very low D29 antibody levels, along with the finding that only 28.6% (8.5%, 48.7%) of estimated VE was mediated by nAb-ID50 titer (Table 4), suggest that: (1) low antibody levels (unquantifiable/undetectable by the immune assays used here) may protect against severe-critical COVID-19; and (2) markers of other immune functions are likely also correlates of protection against severe-critical COVID-19. Moreover, the lower estimated VE against moderate COVID-19 at unquantifiable nAb-ID50 titer, 32.5% (14.8% at negative Spike IgG response), is consistent with the idea that T-cell responses play an important role in preventing severe disease even at low antibody levels[10]. In support of this hypothesis, CD8 + T-cell count was shown to associate with survival in patients with both COVID-19 and hematologic cancer (and hence impaired humoral immunity)[11]. Other studies have also provided evidence that T cells may play a role in preventing severe COVID-19: both the magnitude and frequency of Spike-specific CD4 + T-cell responses measured in the acute phase of COVID-19 were shown to correlate inversely with disease severity, as did CD4 + T-cell response polyantigenicity[12]. Moreover, SARS-CoV-2-specific CD4 + T-cell response magnitude and SARS-CoV-2-specific

CD8 + T-cell response magnitude were each inversely associated with peak disease severity in a cohort of consisting of patients with acute COVID-19 and convalescent donors[13]. Mechanistic insight into the beneficial role of CD8 + T cells against severe disease was provided by Peng et al., who reported that NP$_{105-113}$-B*07:02-specific CD8$^+$ T cell response magnitude associated inversely with disease severity and that NP$_{105-113}$-B*07:02-specific CD8$^+$ T cells showed a highly diverse TCR repertoire, high functional avidity, and antiviral activity as measured by suppression of SARS-CoV-2 replication[14].

It is also possible that non-neutralizing Fc effector functions contribute to protection against severe COVID-19[15]. Although relatively little data are currently available to support this hypothesis, passively administered non-neutralizing antibodies were shown to confer protection against severe disease in a mouse model of SARS-CoV-2 infection, and this protection was linked to their Fc effector functions[16].

Our findings based on individual-level correlates of protection analysis are consistent with those of previous studies using complementary approaches to investigate correlates of protection against severe COVID-19 disease outcomes. Khoury et al. took a population-level modeling approach including data from seven phase 3 COVID-19 vaccine efficacy trials and one convalescent cohort and reported that a given nAb level (expressed as fold of convalescent, due to assay differences across the studies) predicts higher VE against severe vs. symptomatic COVID-19, with this difference greatest at lowest nAb levels (Fig. 3a in ref. [17]). For example, the nAb level associated with 50% VE against severe COVID-19 was approximately sixfold lower than that associated with 50% VE against symptomatic COVID-19. Subsequently, Cromer et al. applied the model developed by Khoury et al. to show that the prediction for a given nAb level of higher VE against severe vs. symptomatic COVID-19 was most apparent for Ancestral COVID-19, but also held for Alpha, Delta, and Beta COVID-19 (Fig. 3b vs. 3a in ref. [18]). Cromer et al. also validated the model of Khoury et al. with vaccine efficacy/effectiveness estimates from one phase 3 randomized, controlled COVID-19 vaccine trial, seven test-negative design studies, and six retrospective cohort studies, showing significant correlation between the predicted vs. reported vaccine efficacy/effectiveness estimates against severe (Fig. 3B) and against symptomatic (Fig. 3A) COVID-19[19]. A limitation of the model used in refs. [17–19] is its assumption that nAbs alone are responsible for protection against severe disease, and thus potential contributions of T-cell responses or other non-neutralizing functions are not considered.

This study examined whether post-vaccination antibody levels correlated with a severe COVID-19 disease outcome, as well as whether they associated with vaccine protection against the same severe outcome. A strength is that the study analyzed individual-level data from a phase 3 randomized, placebo-controlled efficacy trial (RCT), considered "gold standard" data[20] due to the lack of biases and confounding that can be present in other types of studies. Another strength of the study, given that all data are from the same phase 3 study, is that all severe-critical cases included in the analysis met the same definition for severe COVID-19 disease. Furthermore, multiple distinct statistical frameworks[8] were applied to assess the D29 markers as CoPs of severe-critical COVID-19 (two frameworks used) and against moderate to severe-critical COVID-19 (three frameworks used), with the stochastic interventional vaccine efficacy (SVE) framework not previously applied to assess immune markers as CoPs in the ENSEMBLE trial. Moreover, this analysis of the ENSEMBLE trial assessed current marker levels (as opposed to marker levels measured on a given day post-vaccination, as in our previous correlates analysis of ENSEMBLE) as correlates of instantaneous COVID-19 outcomes, including a severe outcome. Limitations of the present analysis include the fact that the follow-up period considered here was in the pre-Omicron era such that it is unknown whether the antibody measurements analyzed here have the same statistical relationship with VE against Omicron COVID-19, or whether antibody measurements

against e.g. Omicron SARS-CoV-2 would be better correlates of Omicron COVID-19. Another limitation is that the analysis was done in baseline SARS-CoV-2 seronegative participants, whereas the majority of the global population is now SARS-CoV-2 seropositive[21].

Together, the results of this work build evidence for neutralizing antibody titer in particular as a surrogate endpoint for adenovirus-based COVID-19 vaccine protection from severe COVID-19. The analyses support that a lower nAb titer is needed to achieve a high level of vaccine-induced protection against severe-critical COVID-19 than against moderate COVID-19. A potential implication for nAb-endpoint immunobridging studies is that lower nAb titers may be able to be used to infer effectiveness against a severe-critical endpoint than would be required to infer effectiveness against a moderate or "any-severity" endpoint.

## Methods
### Ethics statement
All participants provided written informed consent before enrollment. The ENSEMBLE trial (NCT04505722) adhered to the principles of the Declaration of Helsinki and to the Good Clinical Practice guidelines of the International Council for Harmonisation. The protocol (available with Sadoff et al.[5]) and all amendments were approved by the relevant local ethics committees and Institutional Review Boards (see the Inclusion and Ethics section below for a comprehensive list) according to local regulations. Site PIs were invited as co-authors according to the enrollments performed in the study, and were given the opportunity for intellectual contribution. All experiments were performed in accordance with the relevant guidelines and regulations.

### Trial design
The ENSEMBLE trial enrolled and randomized 44,325 participants 1:1 to Ad26.COV2.S vaccine or placebo, with enrollment beginning on September 21, 2020[5]. Participants were not compensated for their participation. All participants were naïve to any investigational COVID-19 vaccine at enrollment. Participants were enrolled at sites in Argentina, Brazil, Chile, Colombia, Mexico, Peru, South Africa, and the United States. See Supplementary Fig. 1 for a schematic diagram of the trial, sampling time points, and blinded follow up period.

### Study endpoints and correlates analysis cohort
Moderate, severe-critical, and moderate to severe-critical COVID-19 endpoints were defined as in section 8.1.3.1 of the study protocol of Sadoff et al.[5], except with the differences outlined in the "Trial design, study cohort, COVID primary endpoints and case/non-case definitions" section of Methods in Fong et al.[6]. Moderate, severe-critical, and moderate to severe-critical COVID-19 endpoints starting ≥7 days post D29 and ≥ 28 days post vaccination up to the end of the correlates study period are included. The rationale for starting to count endpoints at 7 days is that the D29 antibody markers in participants diagnosed with a COVID-19 endpoint between 1 and 6 days post D29 might have been influenced by SARS-CoV-2 infection.

As in Fong et al.[6], correlates analyses were performed in baseline SARS-CoV-2 seronegative participants in the per-protocol cohort, with the same definition of per-protocol as in ref. [4] Participants with any evidence of SARS-CoV-2 infection or any right censoring up to 7 days post D29 were excluded. Within this correlates analysis cohort:
- Moderate, severe-critical, and moderate to severe-critical cases were the corresponding COVID-19 disease endpoints occurring in the time frame described above.
- Non-case vaccine recipients were sampled into the immunogenicity subcohort with no evidence of SARS-CoV-2 infection up to the end of the correlates study period, which was up to 220 days post D1 (Latin America analyses) or 140 days post D1 (South Africa, United States) but not later than the data cut-off date of July 9, 2021.

## Laboratory methods

**Pseudovirus neutralization assay.** Neutralizing antibody titers against lentiviral particles pseudotyped with full-length SARS-CoV-2 Spike (index strain MN908947.3 harboring the D614G point mutation) were measured using the PhenoSense SARS CoV-2 Assay (Monogram Biosciences). This assay has been validated to CLIA/CAP standards and a detailed methods paper has been published[22]. Briefly, lentiviral particles were produced by co-transfecting HEK 293 cells[23] [source: Master Cell Bank (LC0027490) established by Monogram Biosciences in 2001] with a plasmid driving expression of Spike (pCXAS-SARS-CoV-2-D614G) and a lentiviral backbone plasmid (F-lucP.CNDOΔU3[24]). The lentiviral vector contains a firefly luciferase reporter gene, such that the SARS-CoV-2 pseudotyped virus expresses firefly luciferase after infection of HEK 293 cells. Luminescence, measured in relative light units, is directly proportional to virus inoculum infectivity.

At 2 days post-transfection, pseudovirus stock was collected, filtered, and frozen at < − 70 °C. Next, pseudovirus was incubated for one hour at 37 °C with 10 serial three-fold dilutions of serum samples (all human serum samples were heat-inactivated at 56 °C for 60 min before assays were run). A suspension of HEK 293 cells that had been transiently transfected 24 h prior to assay day with plasmids driving the expression of the ACE2 receptor and of the TMPRSS2 protease was then added to the serum-virus mixtures (10,000 cells per well), after which plates were incubated for 3 days at 37 °C in 7% $CO_2$. After the addition of Steady Glo reagent (Promega) to each well, followed by a brief incubation, luciferase signal (relative luminescence units) was measured using a Luminoskan luminometer. Neutralization titers represent the inhibitory dilution (ID) of serum samples at which RLUs were reduced by 50% (ID50) compared to virus control wells (no serum wells). Data analysis (inhibition curve fitting and ID50 determinations) was done using Monogram proprietary analysis software. Using the WHO International Standard for anti-SARS-CoV-2 immunoglobulin 20/136[25], assay readouts were converted to standardized International Units (IU50/ml) as described in Fong et al.[6]. Assay limits are given in Supplementary Table 10.

**Solid-phase electrochemiluminescence S-binding IgG immunoassay.** Serum IgG binding antibodies against SARS-CoV-2 Spike or the Spike receptor binding domain (RBD) were measured using a validated solid-phase electrochemiluminescence S-binding IgG immunoassay as described[26]. The assay used custom MSD SECTOR plates precoated by Meso Scale Discovery with Spike, RBD, and BSA (as a control) and was performed with a Beckman Coulter Biomek based automation integration platform. Assay steps included heat inactivation of test serum samples (30 min at 56 °C); blocking of plates for 60 min at room temperature with MSD blocker A solution; plate washing; and addition of reference standard, quality control test sample, and human serum test samples to plates. Each test sera sample was assayed in duplicate (within a run) in an 8-point dilution series. Samples were incubated at room temperature for 4 h with shaking, plates were washed to remove unbound antibodies, and antibodies bound to Spike or to RBD were detected using an MSD SULFO-TAG anti-human IgG detection antibody (Meso Scale Diagnostics, R32AJ-1, goat polyclonal) diluted to 1X from a 200X vendor-provided stock. MSD Discovery Workbench software (version 4.0) was used for analysis. Using the WHO International Standard for anti-SARS-CoV-2 immunoglobulin 20/136[25], assay readouts were converted to standardized binding antibody units (BAU) as described[26]. Assay limits are given in Supplementary Table 10.

## Statistical methods

Plots of variants causing the severe-critical cases over time and by region were done in R (version 4.3.1)[27]. Code for generating these plots is provided in the Supplementary Software 1 file.

Immune correlates analyses were performed as pre-specified in the SAP for the previous ENSEMBLE correlates analyses[6], with updates noted below to accommodate the longer follow-up, to accommodate separate correlates analyses against the three study endpoints moderate COVID-19, severe-critical COVID-19, and moderate to severe-critical COVID-19, and to include correlates of protection analyses using the stochastic interventional vaccine efficacy framework as well as to assess exposure-proximal correlates of risk.

**Covariate adjustment.** All analyses adjusted for a baseline risk score as described in ref. 6 The geographic region-pooled analyses (Latin America, South Africa, United States) additionally adjusted for geographic region.

**Correlates of risk in vaccine recipients.** As in Fong et al.[6], the covariate-adjusted hazard ratio of the relevant COVID-19 disease endpoint (moderate, severe-critical, or moderate to severe-critical), across marker tertiles, per 10-fold increase in quantitative marker, or per standard deviation-increase in quantitative marker, was estimated using inverse probability sampling-weighted Cox regression models with 95% confidence and Wald-based p-values. Cox model fits were done in the R package survey (version 4.0)[28], with 95% CIs calculated using the percentile bootstrap. For the plots of marker-conditional cumulative incidence of the relevant COVID-19 disease endpoint, Cox model fits were implemented with the R package vaccine (version 1.2.1)[29] with 95% CIs calculated analytically. Nonparametric dose-response regression was also performed with influence-function-based, Wald-based 95% CIs[30]. Point estimates and 95% CIs for marker-threshold-conditional cumulative incidence were calculated using nonparametric targeted minimum loss-based regression[31].

## Correlate of protection analyses

**Controlled vaccine efficacy.** Vaccine efficacy by level of each antibody marker was estimated using a Cox proportional hazards implementation [done using the R package vaccine (version 1.2.1)[29]] as well as a nonparametric monotone dose-response implementation of the controlled effects approach of Gilbert et al[30]. and Kenny[32] [implemented in the R package vaccine (version 1.2.1)[29]]. The approach estimates the causal parameter of one minus the probability of the relevant COVID-19 disease endpoint (moderate, severe-critical, or moderate to severe-critical) by 181, 170, or 181 days (all for the geographic region-pooled analysis), respectively, post Day 29 under a hypothetical assignment of all participants to receive the vaccine and to have their D29 marker set to a certain specified value, divided by this probability under a hypothetical assignment of all participants to receive the placebo (see the SAP for the previous ENSEMBLE correlates analyses[6], for further details).

**Controlled vaccine efficacy sensitivity analysis.** To assess the robustness of the controlled vaccine efficacy results to potential unmeasured confounders, a sensitivity analysis was conducted. This analysis repeated the analyses noted above assuming the existence of a specified unmeasured confounder that would make it harder to infer a correlate of protection (see the SAP for the previous ENSEMBLE correlates analyses[6] for further details). A sensitivity analysis was also performed based on E-values[33] of the vaccine recipient antibody tertile subgroups. See the Supplementary Appendix of ref. 26 and the SAP for the previous ENSEMBLE correlates analyses[6] for further details.

**Mediation analysis.** Each antibody marker was assessed as a mediator of vaccine efficacy using the methods of Kenny 2023[32], as implemented in the R package vaccine (version 1.2.1)[29]. With this method, interest lies in the proportion of the vaccine-induced risk reduction that is mediated through a given marker at D29. This proportion is defined as

$1 - \frac{\log(NDE)}{\log(RR)}$, where NDE is the natural direct effect of the vaccine (i.e., the risk ratio of the vaccine group with the antibody marker set to the value if assigned placebo relative to the placebo group, the Non-marker mediated VE) and RR is the risk ratio of the vaccine group relative to the placebo group (RR = 1 – Overall VE). Nonparametric estimators are used to estimate the NDE and the RR, which are in turn used to estimate the proportion mediated. From the proportion mediated formula, it is evident that the proportion is not a true proportion, in that it can take values outside of the interval [0, 1]. That is, given RR < 1, NDE < RR implies the proportion mediated is negative (i.e., greater Non-marker mediated VE than Overall VE implies negative proportion mediated).

**Stochastic interventional VE analysis.** The USG COVID-19 Response Team / CoVPN Vaccine Efficacy Trial Immune Correlates Statistical Analysis Plan[34] describes the use of counterfactual risk and VE measures for hypothetical (analyst-specified) changes to the observed distributions of a fixed set of candidate immune correlates of protection; this has been termed a stochastic-interventional (risk or) vaccine efficacy (SVE) for its relation to causal inference parameters that may be defined based on stochastic interventions[35] or modified treatment policies[36].

We estimated the counterfactual mean probability of moderate to severe-critical COVID-19 by 181 days post D29 under posited mean shifts in the measured D29 Spike IgG, RBD IgG, and nAb-ID50 levels. For each D29 marker, measured levels were hypothetically shifted along a grid {0, 0.2, 0.4, 0.6, 0.8, 1}, given on the $\log_{10}$ scale such that −1.0 represents a 10-fold decrease in nAb-ID50 titer or concentration and 1.0 represents a 10-fold increase in its titer or IgG concentration, allowing for stochastic-interventional risk of the moderate to severe-critical COVID-19 endpoint to be evaluated via the techniques of Hejazi et al.[9] and then translated to the VE scale as detailed in the CoVPN Immune Correlates SAP[34]. When hypothetical values of such shifts resulted in more than 10% of participants' counterfactual marker values being placed below an assay's lower limit of detection (LLOD) (for nAb-ID50), or positivity cut-off (for Spike IgG and RBD IgG), the corresponding hypothetical shifts were omitted from the grid. For the grid of shifts considered for analysis, the trajectory of the estimates of stochastic-interventional (risk or) VE along shifts in GM titer or concentration was summarized by a nonparametric working marginal structural model, resulting in a summary measure of the predicted impact of D29 Spike IgG, RBD IgG, or nAb-ID50 levels on VE. Based on the slope of this linear working model, a hypothesis test for whether VE as a function of shifts in GM titer or concentration changes with the shifts is performed, where a small 2-sided p-value supports a correlate of protection.

Code for conducting the stochastic interventional vaccine efficacy analysis is available in the Supplementary Software 2 file. Analyses were implemented using the txshift (version 0.3.8)[37,38] and sl3 (version 1.4.6) packages[39] for the R language and environment for statistical computing[40]; these methods were previously applied to the Moderna COVE vaccine efficacy trial[41,42].

**Antibody decay and Cox modeling for exposure-proximal correlates.** For exposure-proximal immune correlates analyses, a regression calibration[43] based approach was adopted as described in Huang and Follmann[44]. A hazards model was considered for time to event (moderate, severe-critical, or moderate to severe-critical COVID-19):

$$\lambda(s) = \lambda_0(s) \exp(Z[\beta_0 + \beta_1 X\{s\}] + \beta_2 W) I(\tau < s), \tag{1}$$

where $s$ is calendar time, $\tau$ is study entry time, $Z$ is treatment indicator (0 and 1 for the placebo and vaccine arm, respectively), $X(s)$ is the true underlying immune marker at calendar time $s$ had the immunoassay been performed on a sample collected on that day, and $W$ are the

baseline covariates baseline risk score and region (for analyses pooling over regions). For each immune marker (Spike IgG, RBD IgG, nAb-ID50), a linear mixed effects model was used to model the immune marker trajectory over time, with fixed effect for time since D29, age, sex and random intercept for individuals, adjusting for the case-control sampling weights. As these analyses restricted to data during the blinded phase, the linear mixed effects model with random intercepts is based on two time points: D29 and D71 (see Supplementary Fig. 57 for marker trajectory from a random sample). Only participants with both D29 and D71 measurements contributed to model fitting [paired D29, D71 measurements from n = 719 participants were used to fit the bAb (Spike, RBD) trajectory models and from n = 259 participants were used to fit the nAb-ID50 trajectory model]. Based on the linear mixed effects model fit, the expected value of the immune marker at every day post D29 was estimated conditional on age, sex, and observed history of immune response measures. Cox model parameters were estimated by maximizing the partial likelihood based on the induced hazard[43]. The instantaneous-hazard vaccine efficacy curve conditional on the immune marker taking current value x, $VE(x) = 1 - \exp(\beta_0 + \beta_1 x)$, was then estimated based on the $\beta$ estimates. The nonparametric bootstrap with 500 samples was used to construct the 95% pointwise confidence interval for $VE(x)$.

**Scorecard for ranking antibody marker immune correlate performance.** As we have previously done[7], we systematically ranked the antibody markers as correlates of the different COVID-19 endpoints. Our comparison was based on three categories of correlate-quality criteria: (1) CoR, (2) CoP−VE modification, and (3) CoP−VE mediation. The two criteria for ranking within category (1) were: the HR point estimate per 10-fold increase and the HR point estimate for High vs. Low tertile. The three criteria for ranking within category (2) were: the fold-increase on the multiplicative scale of controlled VE at unquantifiable/undetectable marker level to the 90th percentile of the marker as estimated using a Cox model, the same increase as estimated using a nonparametric model, and the E-value for the point estimate based on the marginalized Cox model (High vs. Low). The criterion for ranking within category (3) was the point estimate of the proportion of VE mediated through the antibody marker, and for comparing antibody markers for a given COVID-19 endpoint, the lower 95% confidence limit for the proportion of VE mediated was used as a second criterion.

### Reporting summary
Further information on research design is available in the Nature Portfolio Reporting Summary linked to this article.

## Data availability
The data sharing policy of Janssen Pharmaceutical Companies of Johnson & Johnson is available at https://www.janssen.com/clinical-trials/transparency. The data needed to execute the custom code for the immune correlates analyses are proprietary to Janssen and may be obtained from the authors upon reasonable request as determined by an agreement with Yale Open Data Access [YODA] Project through the site http://yoda.yale.edu. Source data for Figs. 2−5 are provided with this paper. Source data are provided with this paper.

## Code availability
Code for generating Fig. 1 and Supplementary Fig. 3 is provided in the Supplementary Software 1 file. Immune correlates analyses were done reproducibly based on publicly available R scripts[45]. The https://github.com/CoVPN/correlates_reporting2 repository hosts multiple modular workflows for correlates of risk/protection analyses and automated reporting of analytic results. Analysis modules used in the present work, along with the figures/tables they were used to generate, include: cor_graphical (graphical descriptions of correlates of risk: Fig. 2 and Supplementary Figs. 4−7); cor_tabular (tabular descriptions

of correlates of risk: Table 1 and Supplementary Tables 2–5); cor_coxph (Cox proportional hazards modeling of risk: Table 2, Table 3, Fig. 3, Supplementary Tables 11–18, Supplementary Figs. 9-16, Supplementary Figs. 41–48, Supplementary Fig. 58); immuno_graphical (graphical descriptions of immunogenicity data: Supplementary Fig. 8); immuno_tabular (tabular descriptions of immunogenicity data: Supplementary Tables 8, 9); and cor_threshold (nonparametric marker-thresholded correlate of risk modeling: Supplementary Figs. 25–32). A complementary repository that handles the upstream data processing aspects of the analysis workflow is available at https://github.com/CoVPN/correlates_processing. Code and documentation for nonparametric and Cox-based inference for controlled risk and controlled vaccine efficacy curves (Fig. 4, Supplementary Figs. 17-24, Supplementary Figs. 33–40) and for calculating mediation effects for our application with no variability in the post vaccination antibody markers across placebo recipients (Supplementary Tables 19-22) is provided in the vaccine R package (version 1.2.1, https://CRAN.R-project.org/package=vaccine), with scripting code available at https://github.com/Avi-Kenny/VaxCurve. See also ref. 30. Code for conducting the exposure-proximal analysis (Fig. 5, Supplementary Figs. 51-56) is publicly available at https://github.com/yinghuang124/Exposure-Proximal. See also ref. 44. Code for conducting the stochastic interventional vaccine efficacy analysis (Supplementary Figs. 49 and 50) is available in the Supplementary Software 2 file. This code draws heavily on the txshift R package, see https://cran.r-project.org/package=txshift and https://github.com/nhejazi/txshift for more extensive documentation. See also refs. 9,41,46.

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

## Acknowledgements

We thank the participants and site staff for their contributions to the ENSEMBLE trial. We also thank Craig Magaret for generating Fig. 1 and Supplementary Fig. 3. This work was partially funded by: Federal funds from the Administration for Strategic Preparedness and Response, Biomedical Advanced Research and Development Authority, under Contract Nos. HHSO100201700018C with Janssen and 75A50122C00008 with Labcorp-Monogram Biosciences; the National Institutes of Health, National Institute of Allergy and Infectious Diseases (NIAID) under Public Health Service Grants UM1 AI068635 (HVTN SDMC) (P.B.G.), UM1 AI068614 (HVTN LOC) (LC) and R37AI054165 (P.B.G.); the Intramural Research Program of NIAID; and the Scientific Computing Infrastructure at Fred Hutch, under ORIP grant S10OD028685. This work was also supported by Janssen Research and Development, an affiliate of Janssen Vaccines and Prevention, and part of the Janssen pharmaceutical companies of Johnson & Johnson. The content is solely the responsibility of the authors and does not necessarily represent the official views of the National Institutes of Health. The findings and conclusions in this report are those of the authors and do not necessarily represent the views of the Department of Health and Human Services or its components.

## Author contributions

Conceptualization: Y.F., D.B., S.R., D.J.S., A.K., M.Carone, D.F., R.A.K., R.O.D., P.B.G.; Methodology: Y.F., D.B., D.J.S., A.K., Y.H., M.Carone, A.B.M., C.J.P., N.S.H., A.V., P.B.G.; Software: Y.F., D.B., A.K., Y.H., M.Carone, Y.L., C.Y., N.S.H., P.B.G.; Validation: O.H., Y.F., D.B., S.R., D.J.S., A.K., Y.H., Y.L., C.Y., A.K.R., P.B.G.; Formal Analysis: O.H., Y.F., D.B., A.K., Y.H., M.Carone, A.L., Y.L., C.Y., M.J., N.S.H., P.B.G.; Investigation: S.R., D.J.S., I.V.D., G.A.V.R., K.M., L.J., F.C., O.A.-A., M.B., B.F., B.C.L., C.M., M.Naisan, M.Naqvi, S.N., S.O.C., A.M., L.S., M.Castro, J.W., C.J.P., D.N.W., J.S., G.E.G., B.G., P.A.G., L.-G.B., A.H.G., V.G.V., F.S., H.S., M.D., J.G.K., L.C., K.M.N., P.B.G.; Data Curation: O.H., Y.F., D.B., S.R., D.J.S., I.V.D., G.A.V.R., A.K., Y.H., M.C.arone, A.B.M., K.M., L.J., F.C., S.O.C., A.L., Y.L., C.Y., M.J., N.S.H., D.N.W., A.K.R., J.H., C.T., A.V.; Visualization: L.N.C., Y.F., A.K., Y.H., L.Y., C.Y., N.S.H., P.B.G.; Funding Acquisition: L.C., P.B.G.; Resources: A.B.M., C.R.H., K.M., L.J., F.C., C.J.P., J.S., G.E.G., R.A.K., R.O.D., P.B.G.; Project Administration: A.B.M., C.R.H., K.M., L.J., F.C., S.O.C., C.J.P., D.N.W., J.S., G.E.G., A.K.R., M.P.A., J.G.K., L.C., K.M.N., D.F., R.A.K., R.O.D., P.B.G.; Supervision: Y.F., A.B.M., C.J.P., J.S., G.E.G., R.A.K., R.O.D., P.B.G.; Writing—original draft: L.N.C., P.B.G.; Writing—review and editing: All.

## Competing interests

J.H. is an employee of Janssen Vaccines & Prevention BV and has stock and/or stock options in Johnson & Johnson. S.R., I.V.D., and C.T. are employees of Janssen Pharmaceuticals and have stock and/or stock options in Johnson & Johnson. F.S. was an employee of J&J at the time the study was conducted and has stock and/or stock options in Johnson & Johnson. F.S. also has shares in GlaxoSmithKline as compensation for past employment. D.J.S., G.A.V.R., J.S., H.S., and M.D. were employees of Janssen Vaccines & Prevention BV and had stock and/or stock options in Johnson & Johnson at the time the work was conducted. J.S. has patents (US 11,384,122 B2) on invention of the Janssen COVID-19 vaccine. A.V. was an employee of Janssen Pharmaceuticals, had stock and/or stock options in Johnson & Johnson at the time the work was conducted, and had all patent rights (US 11,384,122 B2) transferred to Johnson & Johnson. The remaining authors declare no competing interests.

## Inclusion and Ethics

The COV3001 (ENSEMBLE) study was reviewed and approved by all relevant local ethics committees and Institutional Review Boards, listed below: Argentina: ANMAT - Administración Nacional de Medicamentos, Alimentos y Tecnologia Médica (Capital Federal, La Plata, Ramos Mejia—Buenos Aires; Ciudad Autonoma de Buenos Aires), Comite de Etica Dr Carlos Barclay (Capital Federal, Buenos Aires; Ciudad Autonoma de Buenos Aires), Comision Conjunta de Investigacion en Salud—CCIS (La Plata, Ramos Mejia–Buenos Aires), Comite de Bioetica de Fundacion Huesped (Ciudad Autonoma de Buenos Aires), Comité de Docencia e Investigación DIM Clínica Privada (Ramos Mejia, Buenos Aires), Comité de Ética en Investigación Clínica y Maternidad Suizo Argentina (Ciudad Autonoma de Buenos Aires), Comité de Ética en Investigación de CEMIC (Ciudad Autonoma de Buenos Aires), Comite de Etica en Investigacion DIM Clinica Privada (Ramos Mejia, Buenos Aires), Comite de Etica Hospital Italiano de La Plata (La Plata, Buenos Aires), Comite de Etiica en Investigacion Hospital General de Agudos J.M. Ramos Mejia (Ciudad Autonoma de Buenos Aires), Comitéd ética del Instituto Médico Platense (CEDIMP) (La Plata, Buenos Aires), IBC Fundacion Huesped (Ciudad Autonoma de Buenos Aires), IBC Helios Salud (Ciudad Autonoma de Buenos Aires), IBC Hospital General de Agudos J.M. Ramos Mejia (Ciudad Autonoma de Buenos Aires) Brazil: ANVISA – Agência Nacional de Vigilância Sanitária (Salvador, Bahia; Barretos, Campinas, São Paulo, São Jose Rio Preto, Ribeirão Preto, São Caetano do Sul – São Paulo; Santa Maria, Porto Alegre – Rio Grande do Sul; Natal, Rio Grande do Norte; Para, Pará; Belo Horizonte, Minas Gerais; Rio de Janeiro, Nova Iguaçu – Rio de Janeiro; Curitiba, Paraná; Brasília, Distrito Federal; Campo Grande, Mato Grosso do Sul; Criciúma, Santa Catarina; Cuiabá, Mato Grosso), CONEP - Comissão Nacional de Ética em Pesquisa (Salvador, Bahia; São Paulo, São Paulo; Santa Maria, Rio Grande do Sul; Para, Pará;), CAPPESq – Comissão de Ética de Análise para Projetos de Pesquisa—HCFMUSP (São Paulo, São Paulo), CEP da Faculdade de Medicina de São José do Rio

Preto – FAMERP (São Jose Rio Preto, São Paulo), CEP da Faculdade de Medicina do ABC/SP (São Paulo, São Paulo), CEP da Fundação Pio XII - Hospital do Câncer de Barretos/SP (Barretos, São Paulo), CEP da Liga Norteriograndense Contra o Câncer (Natal, Rio Grande do Norte), CEP da Pontificia Universidade Catolica de Campinas / PUC Campinas (Campinas, São Paulo), CEP da Real Benemérita Associaçao Portuguesa de Beneficência - Hospital São Joaquim (São Paulo, São Paulo), CEP da Santa Casa de Misericórdia de Belo Horizonte (Belo Horizonte, Minas Gerais), CEP da Secretaria Municipal De Saúde do Rio de Janeiro—SMS/RJ (Rio de Janeiro, Rio de Janeiro), CEP da Universidade de São Caetano do Sul (CEP da Universidade de São Caetano do Sul, São Paulo), CEP da Universidade Federal de Mato Grosso do Sul—UFMS (Campo Grande, Mato Grosso do Sul), CEP da Universidade Federal de Minas Gerais (Belo Horizonte, Minas Gerais), CEP do Centro de Referência e Treinamento DST/AIDS (São Paulo, São Paulo), CEP do do INI-Ipec/Fiocruz (Rio de Janeiro, Rio de Janeiro), CEP do Grupo Hospitalar Conceição / RS (Porto Alegre, Rio Grande do Sul), CEP do Hospital das Clínicas da Faculdade de Medicina de Ribeirão Preto/USP (Ribeirão Preto, São Paulo), CEP do Hospital de Clinicas da Universidade Federal do Parana—HCUFPR / PR (Curitiba, Paraná), CEP do Hospital de Clínicas de Porto Alegre/HCPA (Porto Alegre, Rio Grande do Sul), CEP do Hospital Geral de Nova Iguaçu (Nova Iguaçu, Rio do Janeiro), CEP do Hospital Municipal São José (Criciúma, Santa Catarina), CEP do Hospital Pró-Cardíaco/RJ (Rio de Janeiro, Rio de Janeiro), CEP do Hospital Sírio Libanês (São Paulo, Sao Paulo), CEP do Hospital Universitário Júlio Muller / MT (Cuiabá, Mato Grosso), CEP do Hospital Universitário Professor Edgard Santos—UFBA (Salvador, Bahia), CEP do Instituto de Cardiologia do Distrito Federal (Brasília, Distrito Federal), CEP do Instituto de Infectologia Emílio Ribas/SP (São Paulo, Sao Paulo), CEP do Instituto de Saude e Bem Estar da Mulher - ISBEM / SP (São Paulo, Sao Paulo), CEP em Seres Humanos do HFSE - Hospital Federal dos Servidores do Estado (Rio de Janeiro, Rio de Janeiro), CONEP - Comissão Nacional de Ética em Pesquisa (Brasília, Distrito Federal, Salvador, Bahia; Belo Horizonte, Minas Gerais; Cuiabá, Mato Grosso; Campo Grande, Mato Grosso do Sul; Nova Iguaçu, Rio Janeiro—Rio Janeiro; Barretos, Campinas, Sao Jose Rio Preto, São Caetano do Sul, Sao Paulo, Ribeirão Preto—Sao Paulo; Porto Alegre, Rio Grande do Sul; Natal, Rio Grande do Norte; Curitiba, Paraná; Criciúma, Santa Catarina) Chile: Comité de Ética de Investigación en Seres Humanos (Santiago, Region Met), Comité Ético Científico Servicio de Salud Metropolitano Central (Santiago, Region Met), Instituto de Salud Pública de Chile (Santiago, Region Met; Talca, Temuco), Comité Ético-Científico Servicio de Salud Metropolitano Sur Oriente (Talca, Santiago), Comité de Evaluación Ética Científica Servicio de Salud Araucanía Sur Temuco (Temuco), Comité Ético Científico Servicio de Salud Metropolitano Central (Viña del Mar) Colombia: CEI de la Fundación Cardiovascular de Colombia (Floridablanca), Comité de Ética en Investigación Clínica de la Costa (Barranquilla), INVIMA—Instituto Nacional de Vigilancia de Medicamentos y Alimentos (Colombia) (Barranquilla), Comite de Etica en Investigacion de la E.S.E. Hospital Mental de Antioquia (Santa Marta), Comite de Etica en la Investigacion CAIMED (Bogotá), INVIMA—Instituto Nacional de Vigilancia de Medicamentos y Alimentos (Colombia) (Bogotá), Comite Corporativo de Etica en Investigacion de la Fundacion Santa Fe de Bogota (Bogotá), Comité de Ética e Investigación Biomédica de la Fundación Valle del Lili (Cali), Comite de Etica e Investigacion IPS Universitaria (Medellin), Comite de Etica en Investigacion Asustencial Cientifica de Alta Complejidad (Bogotá), Comite de Etica en Investigacion Biomedica de la Corporacion Científica Pediatrica de Cali (Cali), Comité de Ética en Investigación Clínica de la Costa (Barranquilla), Comite de Etica en Investigacion de la E.S.E. Hospital Mental de Antioquia (Barrio Barzal Villavicencio), Comite de Etica en Investigacion del area de la Salud de la Universidad del Norte (Barranquilla), Comite de Etica en Investigacion Medplus Centro de Recuperación Integral S.A.S (Bogotá), Comité de Ética en Investigaciones CEI-FOSCAL (Floridablanca), Comite de Etica en la Investigacion CAIMED (Bogotá), Comite de Etica para Investigacion Clinica(CEIC) de la

Fundacion Centro de Investigacion Clinica CIC (Medellin), Comite de Investigaciones y Etica en Investigaciones Hospital Pablo Tobon Uribe (Medellin), INVIMA–Instituto Nacional de Vigilancia de Medicamentos y Alimentos (Colombia) (Barranquilla, Bogotá), Cali, Floridablanca, Medellin Mexico: CEI del Hospital Civil de Guadalajara Fray Antonio Alcalde (Guadalajara, Jalisco), CEI Hospital La Mision (Tijuana, Baja California Norte), CI del Hospital Civil de Guadalajara Fray Antonio Alcalde (Guadalajara, Jalisco), CI Hospital La Mision (Tijuana, Baja California Norte), Comite de Bioseguridad del Instituto Nacional de Salud Publica (Mexico, Distrito Federal; Cuernavaca, Morelos), Comite de Etica en Investigacion del Instituto Nacional de Salud Publica (Mexico, Distrito Federal; Cuernavaca, Morelos), Comité de Bioseguridad del Hospital La Misión S.A. de C.V. (Tijuana, Baja California Norte; Oaxaca, Oaxaca; Merida, Yucatán; Tijuana, Baja California Norte), Comité de Bioseguridad de la Coordinación de Investigación en Salud (IMSS) (Mexico, Estado de Mexico), Comité de Bioseguridad de Médica Rio Mayo (CLINBOR) (Mexico, Distrito Federal), Comité de Bioseguridad del Hospital Universitario "Dr. José Eleuterio González" (Monterrey, Nuevo León), COFEPRIS (Comisión Federal para la Protección contra Riesgos Sanitarios) (Cuernavaca, Morelos; Mexico, Distrito Federal; Monterrey, Nuevo León; Oaxaca, Oaxaca; Merida, Yucatán), Comite de Etica de la Fac de Med de la UANL y Hospital Universitario "Dr. Jose Eleuterio Gonzalez" (Monterrey, Nuevo León), Comite de Etica en Investigacion de la Unidad de Atencion Medica e Investigacion en Salud S.C. (Merida, Yucatán), Comite de Etica en Investigacion de Medica Rio Mayo S.C. (Mexico, Distrito Federal), Comite de Etica en Investigacion de Oaxaca Site Management Organization, S.C. (Oaxaca, Oaxaca), Comite de Etica en Investigacion del Centro Medico Nacional Siglo XXI (IMSS) (Mexico, Estado do Mexico), Comité de Investigación de la Coordinación de Investigación en Salud (IMSS) (Mexico, Estado do Mexico), Comite de Investigacion de la Unidad de Atencion Medica e Investigacion en Salud S.C. (Merida, Yucatán), Comite de Investigacion de Oaxaca Site Management Organization, S.C. (Oaxaca, Oaxaca), Comité de Investigación del Hospital Universitario José Eleuterio González (Monterrey, Nuevo León), Comite de Investigacion Medica Rio Mayo, S.C. (Mexico, Distrito Federal) Peru: Comite Nacional Transitorio de Etica en Invest. de los Ensayos Clinicos de la enfermedad COVID-19 (Iquitos–Maynas, Loreto; Lima, San Miguel – Lima), INS - Instituto Nacional de Salud (Peru) (Lima, San Miguel – Lima; Callao; Iquitos–Maynas, Loreto) South Africa: Department Agriculture, Forestry and Fisheries (DAFF) (Port Elizabeth, Mthatha – Eastern Cape; Cape Town, Worcester – Western Cape; Durban, Ladysmith, Vulindlela – KwaZulu-Natal; Johannesburg, Pretoria, Mamelodi East, Soweto, Tembisa – Gauteng; Rustenburg, Klerksdorp – North West; Bloemfontein, Free State; Middelburg, Mpumalanga; Dennilton, Limpopo), Pharma Ethics (Port Elizabeth, Eastern Cape; Durban, Ladysmith – KwaZulu-Natal; Cape Town, Western Cape; Pretoria, Mamelodi East, Johannesburg, Tembisa – Gauteng; Rustenburg, Klerksdorp – North West; Bloemfontein, Free State; Middelburg, Mpumalanga; Dennilton, Limpopo), SAHPRA - South African Health Products Regulatory Authority (Port Elizabeth, Mthatha – Eastern Cape; Cape Town, Worcester – Western Cape; Durban, Ladysmith, Vulindlela – KwaZulu-Natal; Johannesburg, Pretoria, Mamelodi East, Soweto, Tembisa – Gauteng; Rustenburg, Klerksdorp – North West; Bloemfontein, Free State; Middelburg, Mpumalanga; Dennilton, Limpopo), WIRB (Mamelodi East, Pretoria – Gauteng; Ladysmith, KwaZulu-Natal; Bloemfontein, Free State; Cape Town, Western Cape; Dennilton, Limpopo), Wits Health Consortium (Soweto, Johannesburg – Gauteng; Ladysmith, KwaZulu-Natal; Mthatha, Eastern Cape), Wits Institutional Biosafety Committee (Soweto, Pretoria, Johannesburg, Tembisa – Gauteng; Rustenburg, Klerksdorp – North West; Mthatha, Eastern Cape), University of Cape Town HREC (Cape Town, Worcester – Western Cape); University of Cape Town Institute of Infectious Disease & Molecular Medicine (Cape Town, Worcester – Western Cape), University of Cape Town Institutional Biosafety Committee (Cape Town, Worcester – Western Cape), SAMRC Human Research Ethics Committee Scientific Review (Durban, KwaZulu-

Natal), Sefako Makgatho University Research Ethics Committee (SMUREC) (Pretoria, Gauteng), University of KwaZulu Natal Institutional Biosafety Committee (Durban, KwaZulu-Natal), University of KwaZulu-Natal Ethics (Durban, Vulindlela – KwaZulu-Natal), University of Stellenbosch Ethics Committee (Cape Town, Western Cape), University of KwaZulu Natal Institutional Biosafety Committee (Vulindlela, KwaZulu-Natal) United States: Advarra IBC (Detroit, MI; Chapel Hill, NC; Boston, MA; Seattle, WA; Winston-Salem, NC; Austin, TX; Peoria, IL; Huntsville, AL; Long Beach, CA; Tucson, AZ), Biomedical Institute of New Mexico - IBC (Albuquerque, NM), Birmingham VA Medical Center - Alabama- IBC (Birmingham, AL), Clinical Biosafety Services (Hollywood, FL), Columbia University IBC (New York, NY), Copernicus Group IRB (Austin, Dallas, Houston, San Antonio – TX; Rochester, New York, Bronx, Binghamton – NY; Hillsborough, Hackensack, Newark, New Brunswick – NJ; West Palm Beach, Coral Gables, Hollywood, Miami, Orlando, Gainesville, Tampa, Hallandale Beach, Pinellas Park, The Villages, Jacksonville, Deland – FL; Fort Worth, Dallas, San Antonio – TX; Norfolk, Charlottesville – VA; Matairie, New Orleans – LA; Nashville, Knoxville, Memphis, Bristol – TN; Cincinnati, Cleveland, Columbus, Akron – OH; Detroit, Ann Arbor, Grand Rapids – MI; Philadelphia, Pittsburgh – PA; Stanford, San Diego, San Francisco, Oakland, Long Beach, Anaheim, Sacramento, West Hollywood – CA, Las Vegas, Reno – NV; Chicago, Peoria – IL; Omaha, NE; Mobile, Birmingham, Huntsville – AL; St Louis, Greer, Kansas City – MO; Boston, MA; Harrisburg, SD; Decatur, Atlanta, Savannah – GA; Baltimore, Rockville, Annapolis – MD; New Haven, Hartford – CT; Chapel Hill, Raleigh, Fayetteville, Charlotte, Durham, Winston-Salem – NC; Indianapolis, Valparaiso, Evansville – IN; Seattle, WA; Aurora, CO; Lexington, Louisville – KY; Murray, West Jordan, Salt Lake City – UT; Phoenix, Tucson, Glendale – AZ; Spartanburg, Columbia, North Charleston, Anderson, Charleston, Mount Pleasant – SC; Portland, Medford, Corvallis – OR; Albuquerque, Gallup – NM; Little Rock, AR; Jackson, MS; Newport News, VA, Minneapolis, MN; Lenexa, KS), WIRB (Hackensack, NJ; Dallas, TX; Baltimore, MD; Chicago, IL; Aurora, CO; Winston-Salem, NC; Minneapolis, MN; Orlando, Miami, Gainesville – FL; Philadelphia, Pittsburgh – PA; Boston, MA; St Louis, MO; Bronx, New York, NY; New Brunswick, NJ; Phoenix, AZ; Birmingham, AL; Louisville, KY; Albuquerque, NM; New Orleans, LA; Baltimore, MD; San Francisco, CA; Tampa, FL; Aurora, CO; Columbia, SC; Decatur, GA; Reno, NV; Raleigh, NC; Little Rock, AS), Clinical Biosafety Services (Dallas, San Antonio – TX; San Diego, CA; Lexington, KY; Murray, UT; Greer, Kansas City, St Louis – MO; Rockville, MD; Las Vegas, NV; Cincinnati, Columbus, Akron – OH; Phoenix, Tucson, Glendale – AZ; North Charleston, Anderson – SC; Orlando, Pinellas Park, The Villages, Miami – FL; Birmingham, AL; Valparaiso, Evansville – IN; Lenexa, KS), Columbia University IBC (Bronx, New York), Durham VA Medical Center-IBC (Raleigh, NC), Emory University IRB (Decatur, GA), Environmental Health and Safety Office (Atlanta, GA), Institutional Biosafety Committee (New Orleans, LA), James A. Haley Veterans Hospital_IBC (Tampa, FL), Jesse Brown VA Medical Center- IBC (Chicago, IL), Mass General Brigham IBC (Boston, MA), Mount Sinai- Icahn School of Medicine IBC (New York, NY), New York Blood Center IBC (New York, NY), OHSU IBC (Portland, OR), Partners Institutional Biosafety Committee (Boston, MA), Rocky Mountain Regional VA Medical Center-IBC (Aurora, CO), Rush University Medical Center (Chicago, IL), Rush University Medical Center IBC (Chicago, IL), Rutgers Institutional Biosafety Committee (New Brunswick, NJ), Saint Louis University IBC (St Louis, MO), Saint Michael's Medical Center IRB (Newark, NJ), Southeast Louisiana Veterans Health Care System IBC (New Orleans, LA), St. Jude Children's Research Hospital IBC Committee (Memphis, TN), St. Jude Children's Research Hospital IRB (Memphis, TN), Stanford University Administrative Panel on Human Subjects in Medical Research (Stanford, CA), Temple University–IBC (Philadelphia, PA), The University of Chicago Institutional Biosafety Committee (Chicago, IL), UAMS IBC (Little Rock, AS), UIC IBC (Chicago, IL), University of Alabama at Birmingham Institutional Biosafety Committee (Birmingham, AL), University of Arkansas IRB (Little Rock, AS), University of Kentucky Biological Safety (Lexington, KY), University of Kentucky IRB (Lexington, KY), University of Louisville IRB (Louisville, KY), University of Miami-IBC (Miami, FL), University of Mississippi Medical Center IRB (Jackson, MI), University of Pennsylvania Institutional Biosafety Committee (Philadelphia, PA), University of Pittsburgh IBC (Pittsburgh, Pennsylvania), University of South Florida IRB (Tampa, FL), University of Utah Institutional Biosafety Committee (Salt Lake City, UT), University of Utah IRB (Salt Lake City, UT), UTHealth – IBC (Houston, TX), VA Baltimore Research & Education Foundation (BREF)- IBC (Baltimore, MD), VA Central Arkansas Veterans Healthcare System-IBC (Little Rock, AS), VA James J. Peters Department of VA Medical Center-IBC (Bronx, NY), VA Medical Center - Atlanta-IBC (Decatur, GA), VA Medical Center San Francisco- IBC (San Francisco, CA), VA North Florida/South Georgia IBC (Gainesville, FL), VA North Texas Health Care System IBC (Dallas, TX), VA San Diego Healthcare System IBC (Phoenix, AZ), VA Sierra Nevada Health Care System-IBC (Reno, NV), Vanderbilt University Instituitional Review Board (Nashville, TN), Washington University IBC (St Louis, MO), WCG IBCS (Houston, TX; Orlando, FL), Western Institutional Review Board (San Diego, CA; Detroit, MI; New Orleans, LA; New York, NY), WIRB - IBCS Services (Chicago, IL; New Orleans, LA; Oakland, CA; Minneapolis, MN; Columbus, OH; Lexington, KY), WJB Dorne VA Medical Center IBC (Columbia, SC).

## Additional information

Lindsay N. Carpp [1,33], Ollivier Hyrien[1,2,33], Youyi Fong[1,2,33], David Benkeser [3], Sanne Roels[4], Daniel J. Stieh [5,23,33], Ilse Van Dromme[4], Griet A. Van Roey[5], Avi Kenny[6,24,25], Ying Huang [1,2,6], Marco Carone [6], Adrian B. McDermott [7,26], Christopher R. Houchens[8], Karen Martins[8], Lakshmi Jayashankar[8], Flora Castellino[8], Obrimpong Amoa-Awua[7], Manjula Basappa[7], Britta Flach[7], Bob C. Lin[7], Christopher Moore[7], Mursal Naisan[7], Muhammed Naqvi[7], Sandeep Narpala[7], Sarah O'Connell[7], Allen Mueller[7], Leo Serebryannyy[7], Mike Castro [7], Jennifer Wang[7], Christos J. Petropoulos [9], Alex Luedtke[10], Yiwen Lu[1], Chenchen Yu[1], Michal Juraska [1], Nima S. Hejazi [1,11], Daniel N. Wolfe[8], Jerald Sadoff[5,27], Glenda E. Gray[12,13], Beatriz Grinsztejn[14], Paul A. Goepfert [15], Linda-Gail Bekker [16,17,18], Aditya H. Gaur [19], Valdilea G. Veloso[14], April K. Randhawa [1], Michele P. Andrasik[1], Jenny Hendriks[5], Carla Truyers[4], An Vandebosch[4,28], Frank Struyf [4,29], Hanneke Schuitemaker[5,30], Macaya Douoguih[5,31], James G. Kublin[1], Lawrence Corey [1,20], Kathleen M. Neuzil[21,32], Dean Follmann [22], Richard A. Koup[7], Ruben O. Donis [8] & Peter B. Gilbert [1,2,6 ✉], On behalf of the Immune Assays Team, the Coronavirus Vaccine Prevention Network (CoVPN)/ENSEMBLE Team, the United States Government (USG)/CoVPN Biostatistics Team

[1]Vaccine and Infectious Disease Division, Fred Hutchinson Cancer Center, Seattle, WA, USA. [2]Public Health Sciences Division, Fred Hutchinson Cancer Center, Seattle, WA, USA. [3]Department of Biostatistics and Bioinformatics, Rollins School of Public Health, Emory University, Atlanta, GA, USA. [4]Johnson & Johnson Innovative Medicine, Beerse, Belgium. [5]Janssen Vaccines and Prevention, Leiden, the Netherlands. [6]Department of Biostatistics, University of Washington School of Public Health, Seattle, WA, USA. [7]Vaccine Research Center, National Institute of Allergy and Infectious Diseases, National Institutes of Health, Bethesda, MD, USA. [8]Biomedical Advanced Research and Development Authority, Washington, DC, USA. [9]LabCorp-Monogram Biosciences, South San Francisco, CA, USA. [10]Department of Statistics, University of Washington, Seattle, WA, USA. [11]Department of Biostatistics, T.H. Chan School of Public Health, Harvard University, Boston, MA, USA. [12]Perinatal HIV Research Unit, Faculty of Health Sciences, University of the Witwatersrand, Johannesburg, South Africa. [13]South African Medical Research Council, Cape Town, South Africa. [14]Evandro Chagas National Institute of Infectious Diseases-Fundação Oswaldo Cruz, Rio de Janeiro, Brazil. [15]Division of Infectious Diseases, Department of Medicine, University of Alabama at Birmingham, Birmingham, AL, USA. [16]Institute of Infectious Disease and Molecular Medicine, University of Cape Town, Observatory, Cape Town, South Africa. [17]Department of Medicine, University of Cape Town and Groote Schuur Hospital, Observatory, Cape Town, South Africa. [18]Desmond Tutu HIV Centre, University of Cape Town, Cape Town, South Africa. [19]Department of Infectious Diseases, St. Jude Children's Research Hospital, Memphis, TN, USA. [20]Department of Laboratory Medicine and Pathology, University of Washington, Seattle, WA, USA. [21]Center for Vaccine Development and Global Health, University of Maryland School of Medicine, Baltimore, MD, USA. [22]Biostatistics Research Branch, National Institute of Allergy and Infectious Diseases, National Institutes of Health, Bethesda, MD, USA. [23]Present address: Vaccine Company Inc. , South San Francisco, CA, USA. [24]Present address: Department of Biostatistics and Bioinformatics, Duke University, Durham, NC, USA. [25]Present address: Duke Global Health Institute, Duke University, Durham, NC, USA. [26]Present address: Sanofi Vaccines R&D, Marcy l'étoile, France. [27]Present address: Centivax, South San Francisco, CA, USA. [28]Present address: argenx BV, Ghent, Belgium. [29]Present address: GSK, Wavre, Belgium. [30]Present address: Valneva, Saint-Herblain, France. [31]Present address: Merck, Rahway, NJ, USA. [32]Present address: Fogarty International Center, Bethesda, MD, USA. [33]These authors contributed equally: Lindsay N. Carpp, Ollivier Hyrien, Youyi Fong. ✉e-mail: pgilbert@fredhutch.org

## the Immune Assays Team

Adrian B. McDermott [7,26], Christopher R. Houchens[8], Karen Martins[8], Lakshmi Jayashankar[8], Flora Castellino[8], Obrimpong Amoa-Awua[7], Manjula Basappa[7], Britta Flach[7], Bob C. Lin[7], Christopher Moore[7], Mursal Naisan[7], Muhammed Naqvi[7], Sandeep Narpala[7], Sarah O'Connell[7], Allen Mueller[7], Leo Serebryannyy[7], Mike Castro [7], Jennifer Wang[7], Christos J. Petropoulos [9], Daniel N. Wolfe[8], Richard A. Koup[7] & Ruben O. Donis [8]

## the Coronavirus Vaccine Prevention Network (CoVPN)/ENSEMBLE Team

Ilse Van Dromme[4], Griet A. Van Roey[5], Jerald Sadoff[5,27], Glenda E. Gray[12,13], Beatriz Grinsztejn[14], Paul A. Goepfert [15], Linda-Gail Bekker [16,17,18], Aditya H. Gaur [19], Jenny Hendriks[5], Carla Truyers[4], An Vandebosch[4,28], Frank Struyf [4,29], Hanneke Schuitemaker[5,30], Macaya Douoguih[5,31], James G. Kublin[1], Lawrence Corey [1,20] & Kathleen M. Neuzil[21,32]

## the United States Government (USG)/CoVPN Biostatistics Team

Lindsay N. Carpp[1,33], Ollivier Hyrien[1,2,33], Youyi Fong[1,2,33], David Benkeser [3], Avi Kenny[6,24,25], Ying Huang [1,2,6], Marco Carone [6], Alex Luedtke[10], Yiwen Lu[1], Chenchen Yu[1], Michal Juraska [1], Nima S. Hejazi [1,11], April K. Randhawa [1], Dean Follmann [22] & Peter B. Gilbert [1,2,6 ✉]

