## [Peer Review File · Nature Communications]

Neutralizing Antibody Correlate of Protection Against Severe-Critical COVID-19 in the ENSEMBLE Single-Dose Ad26.COVS Vaccine Efficacy TrialREVIEWER COMMENTS

Reviewer #1 (Remarks to the Author):

The authors assessed the relationship between antibody levels post-vaccination and the risk and protection against severe-critical COVID-19 (correlates of protection, CoP). The study utilises data from the ENSEMBLE trial, focusing on binding and neutralizing antibody markers as correlates of risk (CoR) and CoP against severe-critical COVID-19. To my knowledge, there was no previous publication on CoP against severe outcomes using clinical trial data, and therefore this study provides a valuable assessment on severe-critical COVID-19 CoP.

This team has published a few studies on COVID vaccine CoP on different vaccine platforms. The statistical methodologies in this paper, following previous studies by the same team, are comprehensive and robust, incorporating a range of methods such as CoR, CoP on VE modification and mediation analysis. These analyses provide consistent findings. The paper's conclusion, suggesting not too much antibody levels may suffice for protection against severe COVID-19 outcomes and highlighting the potential role of other immune markers such as T cells, aligns with findings from other studies using varied designs. However, a more extensive discussion comparing these results with existing literature would provide a richer context. Additionally, incorporating discussion on biological mechanisms to support the findings would benefit wider readers, such as researchers in immunology.

Minor comments:

1. Line 108. The authors mentioned Reference as a variant type. I searched in supp and found it refers to the index strain (GenBank accession number: MN908947.3) harboring the D614G point mutation. Would be helpful this can be added to the main for clearer understanding.
2. Lines 133-139 – Titer Selection: Clarification is needed on how specific nAR-ID50 titer values (e.g., 5.3, 12.3, 5.7 IU/ml) were determined for risk estimation. Are these based on antibody level tertiles or another criteria?

3. Considering the multiple statistical methods applied, sharing the code (possibly on GitHub) would enhance understanding and transparency and allow for replication and further analysis on vaccine CoP.

Reviewer #2 (Remarks to the Author):

In this paper, Carpp et al analyze the level of seroneutralization at week 4 in the Ensemble vaccine trial (Ad26.COVS.S). They discuss the value of neutralization as a CoP against both severe and moderate disease. The study adds to the initial evaluation of the same data by Fong et al (ref 5) with data analyzed up to 6 months after injection (as compared to 3 months in the initial study). However using long term data also would probably require different methodologies, taking into account the time dependent effects: waning immunity to adjust the level of antibodies at the time of exposure, change in circulating strains that may affect neutralization, not to mention other CoP (see, among many other studies Seekircher et al, Lancet Microbe 2023 ; Hertz et al, Nature Comm 2023). The finding that lower nAbs are sufficient to protect against severe disease than against symptomatic covid (the main endpoint used in studies) is also not new: see discussion in Cromer et al, Lancet Microbe 2022; or simply the fact that first generation vaccine continued to be effective against severe disease while they barely neutralize Omicron variants.

So overall, the finding and the methodology that neutralizing antibody are associated with symptomatic or even severe disease is not new, and the specific finding of this study are relatively outdated (vaccine no longer used, entirely pre-omicron, in a fully naïve population), making it difficult to be used for operational purposes.

Response to reviewer comments:

Reviewer #1 (Remarks to the Author):

The authors assessed the relationship between antibody levels post-vaccination and the risk and protection against severe-critical COVID-19 (correlates of protection, CoP). The study utilises data from the ENSEMBLE trial, focusing on binding and neutralizing antibody markers as correlates of risk (CoR) and CoP against severe-critical COVID-19. To my knowledge, there was no previous publication on CoP against severe outcomes using clinical trial data, and therefore this study provides a valuable assessment on severe-critical COVID-19 CoP.

Response: Thank you for the positive comments.

This team has published a few studies on COVID vaccine CoP on different vaccine platforms. The statistical methodologies in this paper, following previous studies by the same team, are comprehensive and robust, incorporating a range of methods such as CoR, CoP on VE modification and mediation analysis. These analyses provide consistent findings. The paper's conclusion, suggesting not too much antibody levels may suffice for protection against severe COVID-19 outcomes and highlighting the potential role of other immune markers such as T cells, aligns with findings from other studies using varied designs. However, a more extensive discussion comparing these results with existing literature would provide a richer context.

Response: We have expanded our Discussion to include a comparison of our work with the present literature on correlates of severe COVID-19:

“Our findings based on individual-level correlates of protection analysis are consistent with those of previous studies using complementary approaches to investigate correlates of protection against severe COVID-19 disease outcomes. Khoury et al. took a population-level modeling approach including data from seven phase 3 COVID-19 vaccine efficacy trials and one convalescent cohort and reported that a given nAb level (expressed as fold of convalescent, due to assay differences across the studies) predicts higher VE against severe vs. symptomatic COVID-19, with this difference greatest at lowest nAb levels (Fig. 3a in ref.¹). For example, the nAb level associated with 50% VE against severe COVID-19 was approximately six-fold lower than that associated with 50% VE against symptomatic COVID-19. Subsequently, Cromer et al. applied the model developed by Khoury et al. to show that the prediction for a given nAb level of higher VE against severe vs. symptomatic COVID-19 was most apparent for Ancestral COVID-19, but also held for Alpha, Delta, and Beta COVID-19 (Fig. 3b vs. 3a in ref.²). Cromer et al. also validated the model of Khoury et al. with vaccine efficacy/effectiveness estimates from one phase 3 randomized, controlled COVID-19 vaccine trial, seven test-negative design studies, and six retrospective cohort studies, showing significant correlation between the predicted vs. reported vaccine efficacy/effectiveness estimates against severe (Fig. 3B) and against symptomatic (Fig. 3A) COVID-19.³ A limitation of the model used in refs.¹⁻³ is its assumption that nAbs alone are responsible for protection against severe disease, and thus potential contributions of T-cell responses or other non-neutralizing functions are not considered.” (pp. 11-12)

Additionally, incorporating discussion on biological mechanisms to support the findings would benefit wider readers, such as researchers in immunology.

Response: We have added the following text to the Discussion:

“In support of this hypothesis, CD8+ T-cell count was shown to associate with survival in patients with both COVID-19 and hematologic cancer (and hence impaired humoral immunity).⁴ Other studies have also provided evidence that T cells may play a role in preventing severe COVID-19: both the magnitude and frequency of Spike-specific CD4+ T-cell responses measured in the acute phase of COVID-19 were shown to correlate inversely with disease severity, as did CD4+ T-cell response polyantigenicity.⁵ Moreover, SARS-CoV-2-specific CD4+ T-cell response magnitude and SARS-CoV-2-specific CD8+ T-cell response magnitude were each inversely associated with peak disease severity in a cohort of consisting of patients with acute COVID-19 and convalescent donors.⁶ Mechanistic insight into the beneficial role of CD8+ T cells against severe disease was provided by Peng et al., who reported that NP105–113-B*07:02-specific CD8+ T cell response magnitude associated inversely with disease severity and that NP105–113-B*07:02-specific CD8+ T cells showed a highly diverse TCR repertoire, high functional avidity, and antiviral activity as measured by suppression of SARS-CoV-2 replication.⁷

It is also possible that non-neutralizing Fc effector functions contribute to protection against severe COVID-19.⁸ Although relatively little data are currently available to support this hypothesis, passively administered non-neutralizing antibodies were shown to confer protection against severe disease in a mouse model of SARS-CoV-2 infection, and this protection was linked to their Fc effector functions.^{9”} (pp. 10-11)

Minor comments:

1. Line 108. The authors mentioned Reference as a variant type. I searched in supp and found it refers to the index strain (GenBank accession number: MN908947.3) harboring the D614G point mutation. Would be helpful this can be added to the main for clearer understanding.

Response: We have added this in the paragraph starting “The SARS-CoV-2 variants causing the severe-critical cases varied over time...” (p. 5)

2. Lines 133-139 – Titer Selection: Clarification is needed on how specific nAR-ID50 titer values (e.g., 5.3, 12.3, 5.7 IU/ml) were determined for risk estimation. Are these based on antibody level tertiles or another criteria?

Response: Risk is not estimated only for subgroups of vaccine recipients defined by having marker levels at the quoted values in the text, but rather it was estimated across a range of subgroups of vaccine recipients, defined by having a given marker value, for all values ranging from negative response (for the binding antibody markers) or unquantifiable response (for the neutralizing antibody marker) out to the 90th percentile (for the nonparametrically estimated curve). The choice of the first two specific nAb-ID50 titers at which estimated risk was quoted in the text (unquantifiable, 5.2 IU50/ml) are selected because undetectable is of distinct interest on its own (lowest possible titer value representing a ‘negative response’ subgroup and because 5.2 IU50/ml represents a value just above the LLOQ of 4.8975 AU/ml. We did not have a solid rationale for the

third selected value, so to make the choice more objective, in the revision we have selected the 90th percentile value, because that value had been pre-defined as the highest titer at which estimates would be calculated.

We have made the following revisions (underlined) to the manuscript:

“Cumulative incidence of severe-critical COVID-19 through 170 days post-D29 decreased across the analyzed ranges of vaccine recipient subgroups defined by D29 antibody levels at a specific value. For nAb-ID50, the cumulative incidence of severe-critical COVID-19 was estimated by a nonparametric method over values ranging from unquantifiable titer to the 90th percentile (30.2 IU50/ml).” (p. 6)

We also note that the ranges of marker values over which risk is estimated is included in the captions of Supplementary Figs. 17-24. For the threshold correlate of risk figures (Supplementary Figs. 25-32), the rightmost point of the curve is determined by the number of cases satisfying the threshold condition, which has to be at least 5.

We have also applied a similar philosophy when quoting point estimates of controlled vaccine efficacy (VE) at specific marker values, where VE point estimates at the same nAb-ID50 titers (unquantifiable, just-quantifiable of 5.2 IU50/ml, and 90th percentile 30.2 IU50/ml) are now quoted for severe-critical and for moderate COVID-19 (bottom of p. 7).

Likewise, for Spike IgG bAb, VE point estimates at the same Spike IgG concentrations (negative response, just-positive concentration of 11.1 BAU/ml, and 90th percentile concentration 125 BAU/ml) are now quoted for severe-critical and for moderate COVID-19 (bottom of p. 7).

Also, information on the ranges of marker values over which VE is estimated is included in the captions of Figure 4, Supplementary Figs. 33-48, 51-56, and 58.

3. Considering the multiple statistical methods applied, sharing the code (possibly on GitHub) would enhance understanding and transparency and allow for replication and further analysis on vaccine CoP.

Response: We fully agree, and have added the following Code Availability statement to the revised manuscript:

“Immune correlates analyses were done reproducibly based on publicly available R scripts hosted on the GitHub collaborative programming platform (https://github.com/CoVPN/correlates_reporting2, <https://github.com/Avi-Kenny/VaxCurve>, and <https://github.com/yinghuang124/Exposure-Proximal>). The code used for the stochastic interventional vaccine efficacy analysis is available in the Supplementary Software file. Methodological components of the “VaxCurve” code are publicly available at <https://CRAN.R-project.org/package=vaccine>.”

Reviewer #2 (Remarks to the Author):

In this paper, Carpp et al analyze the level of seroneutralization at week 4 in the Ensemble vaccine trial (Ad26.COVS.S). They discuss the value of neutralization as a CoP against both

severe and moderate disease. The study adds to the initial evaluation of the same data by Fong et al (ref 5) with data analyzed up to 6 months after injection (as compared to 3 months in the initial study). However using long term data also would probably require different methodologies, taking into account the time dependent effects: waning immunity to adjust the level of antibodies at the time of exposure, change in circulating strains that may affect neutralization, not to mention other CoP (see, among many other studies Seekircher et al, Lancet Microbe 2023 ; Hertz et al, Nature Comm 2023).

Response: Thank you for raising the issue that the use of long-term follow up data may require different statistical methodologies for immune correlates assessment. To address this issue, we have added to the revision the results of an “exposure-proximal” immune correlates of protection analysis. In this exposure-proximal analysis, inferences were made for an antibody marker level that is hypothetically available from a serum sample on every single day of follow-up as a correlate of the probability of COVID-19 occurrence over the next day.

In the revision we now state (pp. 9-10): “Given that this analysis assesses immune correlates through ~7 months post-vaccination, whereas our previous correlates analysis of ENSEMBLE⁵ assessed through ~2.5 months post-vaccination, waning of antibody levels over time is important to consider. Using a measurement error statistical method, we performed an exposure-proximal correlates analysis for a hypothetical scenario where the antibody marker under study was repeatedly measured from serum samples collected on every day of follow-up, and the analysis assesses how the current value of this daily measured marker correlates with the hazard of COVID-19 (i.e., the probability of COVID-19 occurrence over the next day) (see Methods for details). From these current-marker conditional hazard curves, we generated current-marker conditional VE curves (exposure-proximal VE) by dividing the conditional hazard curve by the hazard of COVID-19 for the whole placebo arm.”

Note that by “current” we refer to the true underlying antibody marker level not subject to technical measurement error, in a hypothetical scenario in which the value was available from serum samples collected every day over the follow-up period. Figure 5 presents a side-by-side comparison of curves showing estimated exposure-proximal vaccine efficacy against severe-critical COVID-19 and against moderate COVID-19 by current nAb-ID50 level. Analogous panels for current Spike IgG are also shown in Figure 5, with corresponding RBD IgG results provided in Supplementary Figs. 51 and 52.

We state in the revision (p. 10):

“Figure 5a shows that exposure-proximal VE against severe-critical COVID-19 rose as current nAb-ID50 titer increased across the range of analyzed values (unquantifiable titer up to the 97.5th percentile). Similar results were obtained for current Spike IgG concentration, albeit with a less steep increase and with wider 95% CIs, especially at the ends of the curve (Figure 5c). Similarly, exposure-proximal VE against moderate COVID-19 increased with current nAb-ID50 titer (Figure 5b) as well as with current Spike IgG concentration (Figure 5d). Latin America-specific exposure-proximal VE curves against severe-critical COVID-19 are shown in Supplementary Fig. 54 and against moderate COVID-19 in Supplementary Fig. 55; these results, which were similar to those in the pooled analysis, are discussed in the Supplementary Text.”

For completeness, curves showing estimated exposure-proximal vaccine efficacy against moderate to severe-critical COVID-19 by current levels of each of the three markers are shown in Supplementary Fig. 53. Geographic region-specific results were also assessed, where possible. Supplementary Figs. 54 and 55 show curves for estimated exposure-proximal vaccine efficacy against severe-critical COVID-19 or moderate COVID-19, respectively, by current marker level (all three markers) for the Latin America cohort; Supplementary Fig. 56 shows estimated exposure-proximal vaccine efficacy curves (all three markers) for the moderate to severe-critical COVID-19 endpoint, for each of the three geographic regions separately.

Supplementary Text:

“Supplementary Fig. 54 shows the results of assessing each current marker as an exposure-proximal correlate of severe-critical COVID-19 in the Latin America cohort. Estimated exposure-proximal VE against severe-critical COVID-19 rose as current nAb-ID50 titer increased across the range of analyzed values (Supplementary Fig. 54C). Similar results were obtained for the two bAb markers (Supplementary Figs. 54A, B), except that the curves appeared less steep than the nAb-ID50 curve and had substantially wider 95% CIs at the left-end tail of each curve. Supplementary Fig. 55C shows that estimated VE against moderate COVID-19 also increased with current nAb-ID50 titer, with similar results for the binding antibody markers in the severe-critical COVID-19 analysis (somewhat flatter bAb vs. nAb curves, with wider 95% CIs on the bAb curves) (Supplementary Fig. 55A, B).”

The finding that lower nAbs are sufficient to protect against severe disease than against symptomatic covid (the main endpoint used in studies) is also not new: see discussion in Cromer et al, Lancet Microbe 2022; or simply the fact that first generation vaccine continued to be effective against severe disease while they barely neutralize Omicron variants. So overall, the finding and the methodology that neutralizing antibody are associated with symptomatic or even severe disease is not new, and the specific finding of this study are relatively outdated (vaccine no longer used, entirely pre-omicron, in a fully naïve population), making it difficult to be used for operational purposes.

Response: We have expanded the Discussion to place our findings into the context of existing work on correlates of severe COVID-19.

“Our findings based on individual-level correlates of protection analysis are consistent with those of previous studies using complementary approaches to investigate correlates of protection against severe COVID-19 disease outcomes. Khoury et al. took a population-level modeling approach including data from seven phase 3 COVID-19 vaccine efficacy trials and one convalescent cohort and reported that a given nAb level (expressed as fold of convalescent, due to assay differences across the studies) predicts higher VE against severe vs. symptomatic COVID-19, with this difference greatest at lowest nAb levels (Fig. 3a in ref.¹). For example, the nAb level associated with 50% VE against severe COVID-19 was approximately six-fold lower than that associated with 50% VE against symptomatic COVID-19. Subsequently, Cromer et al. applied the model developed by Khoury et al. to show that the prediction for a given nAb level of higher VE against severe vs. symptomatic COVID-19 was most apparent for Ancestral COVID-19, but also held for Alpha, Delta, and Beta COVID-19 (Fig. 3b vs. 3a in ref.²). Cromer et al.

also validated the model of Khoury et al. with vaccine efficacy/effectiveness estimates from one phase 3 randomized, controlled COVID-19 vaccine trial, seven test-negative design studies, and six retrospective cohort studies, showing significant correlation between the predicted vs. reported vaccine efficacy/effectiveness estimates against severe (Fig. 3B) and against symptomatic (Fig. 3A) COVID-19.³ A limitation of the model used in refs.¹⁻³ is its assumption that nAbs alone are responsible for protection against severe disease, and thus potential contributions of T-cell responses or other non-neutralizing functions are not considered.” (pp. 11-12)

We also now include a paragraph describing the strengths and limitations of this work, where we include many of the Reviewer’s points:

“To our knowledge, this is the first analysis of individual-level data from a phase 3 randomized, placebo-controlled efficacy trial (RCT) – considered “gold standard” data¹⁰ due to the lack of biases and confounding that can be present in other types of studies – to examine whether post-vaccination antibody levels correlate with a severe COVID-19 disease outcome, as well as whether they associate with vaccine protection against the same severe outcome. One strength of the study, given that all data are from the same phase 3 study, is that all severe-critical cases included in the analysis met the same definition for severe COVID-19 disease. Another strength of the study is the application of multiple distinct statistical frameworks¹¹ to assess the D29 markers as CoPs of severe-critical COVID-19 (two frameworks used) and against moderate to severe-critical COVID-19 (three frameworks used), with the stochastic interventional vaccine efficacy (SVE) framework not previously applied to assess immune markers as CoPs in the ENSEMBLE trial. Moreover, this is the first analysis of the ENSEMBLE trial to assess current marker levels as correlates of instantaneous COVID-19 outcomes, including a severe outcome. Limitations of the present analysis include the fact that the follow-up period considered here was in the pre-Omicron era such that it is unknown whether the antibody measurements analyzed here have the same statistical relationship with VE against Omicron COVID-19, or whether antibody measurements against e.g. Omicron SARS-CoV-2 would be better correlates of Omicron COVID-19. Another limitation is that the analysis was done in baseline SARS-CoV-2 seronegative participants, whereas the majority of the global population is now SARS-CoV-2 seropositive.^{12”} (p. 12)

Moreover, as to the comment that the methodology is not new, we have added two new supplementary figures (Supplementary Figs. 49, 50) showing results from a distinct statistical framework for assessing correlates of protection, stochastic interventional VE (SVE). As we recently summarized in our recent review [Gilbert et al. 2024 *Vaccine*, “Four statistical frameworks for assessing an immune correlate of protection (surrogate endpoint) from a randomized, controlled, vaccine efficacy trial”], this analysis assesses how overall VE would be expected to change under user-specified shifts d of the marker values of vaccine recipients from their observed values. This framework is distinct from the two other frameworks used in the manuscript (controlled VE and mediation of VE).

Of the five phase 3 COVID-19 vaccine trials for which immune correlates of protection were assessed under the US Government’s COVID-19 Vaccine Correlates of Protection Program,¹³ the SVE method has so far only been applied to Moderna COVE, for which results have been reported.^{14,15} Application of the SVE method to the ENSEMBLE trial

adds to the body of evidence¹³ supporting binding and neutralizing antibodies as correlates from the other methods¹⁶ using a different statistical lens.

The results provided further support for D29 nAb-ID50 titer as a correlate of protection against moderate to severe-critical COVID-19, in both the geographic region-pooled analysis (Supplementary Fig. 49C) and in the Latin America analysis (Supplementary Fig. 50). They also supported D29 RBD IgG binding antibody concentration as a correlate of protection against moderate to severe-critical COVID-19 in the pooled analysis (Supplementary Fig. 49B); evidence for D29 Spike IgG was weaker (Supplementary Fig. 49A). We have added the following to the revision (pp. 9-10):

“We also applied a third statistical framework for assessing CoPs, stochastic interventional VE (SVE),⁸ to assess the D29 markers as CoPs against moderate to severe-critical COVID-19. In this framework, VE is estimated under hypothetical immune marker shifts applied to all individual vaccine recipients, relative to their observed immune marker levels. For D29 nAb-ID50, estimated VE generally increased with successive shifts in titer: At no D29 nAb-ID50 shift, estimated SVE was 47.7% (95% CI: 44.6%, 50.7%), and with 1.6-fold, 4-fold, and 10-fold shifts, estimated SVE was 57.3% (53.4%, 60.8%), 54.4% (47.9%, 60.1%), and 62.9% (54.2%, 69.9%), respectively (Supplementary Fig. 49c). The p-value for testing the hypothesis that VE changes as a function of shift in D29 nAb-ID50 titer (see Methods) was <0.001, providing further evidence in support of D29 nAb-ID50 as a CoP against moderate to severe-critical COVID-19. A similar result was seen for D29 RBD IgG, with estimated SVE increasing to 49.7% (46.0%, 53.1%), 58.5% (51.5%, 64.5%), and 69.7% (54.0%, 80.0%), respectively, at the same shifts of 1.6-fold, 4-fold, and 10-fold in IgG concentration (p=0.007 for testing the hypothesis that VE changes as a function of shift in D29 RBD IgG concentration) (Supplementary Fig. 49b). For D29 Spike IgG, the increases in SVE were smaller with shifted IgG concentration and the p-value for testing the hypothesis that VE changes as a function of shift in D29 Spike IgG concentration was 0.12 (Supplementary Fig. 49a).”

Supplementary Text (p. 17):

“Latin America: Stochastic interventional vaccine efficacy (SVE)

We applied the stochastic-interventional VE (SVE) framework² to assess the D29 markers as CoPs against moderate to severe-critical COVID-19 in the Latin America cohort. For D29 nAb-ID50, estimated VE increased with shifts in titer: At no D29 nAb-ID50 shift, estimated SVE was 38.7% (95% CI: 34.3%, 42.8%), and with 1.6-fold, 4-fold, and 10-fold shifts, estimated SVE increased to 43.8% (37.8%, 49.2%), 54.5% (41.2%, 64.8%), and 59.4% (46.9%, 68.9%), respectively (Supplementary Fig. 50). The p-value for SVE changing with D29 nAb-ID50 was <0.001, providing further evidence in support of D29 nAb-ID50 as a CoP against moderate to severe-critical COVID-19. SVE estimates for the binding antibody markers (Spike IgG, RBD IgG) were not stable and are not shown.”

References

1. Khoury DS, Cromer D, Reynaldi A, et al. Neutralizing antibody levels are highly predictive of immune protection from symptomatic SARS-CoV-2 infection. *Nat Med* 2021; **27**(7): 1205-11.

2. Cromer D, Steain M, Reynaldi A, et al. Neutralising antibody titres as predictors of protection against SARS-CoV-2 variants and the impact of boosting: a meta-analysis. *Lancet Microbe* 2022; **3**(1): e52-e61.
3. Cromer D, Steain M, Reynaldi A, et al. Predicting vaccine effectiveness against severe COVID-19 over time and against variants: a meta-analysis. *Nat Commun* 2023; **14**(1): 1633.
4. Bange EM, Han NA, Wileyto P, et al. CD8(+) T cells contribute to survival in patients with COVID-19 and hematologic cancer. *Nat Med* 2021; **27**(7): 1280-9.
5. Tarke A, Potesta M, Varchetta S, et al. Early and Polyantigenic CD4 T Cell Responses Correlate with Mild Disease in Acute COVID-19 Donors. *Int J Mol Sci* 2022; **23**(13).
6. Rydzynski Moderbacher C, Ramirez SI, Dan JM, et al. Antigen-Specific Adaptive Immunity to SARS-CoV-2 in Acute COVID-19 and Associations with Age and Disease Severity. *Cell* 2020; **183**(4): 996-1012 e19.
7. Peng Y, Felce SL, Dong D, et al. An immunodominant NP(105-113)-B*07:02 cytotoxic T cell response controls viral replication and is associated with less severe COVID-19 disease. *Nat Immunol* 2022; **23**(1): 50-61.
8. Zhang A, Stacey HD, D'Agostino MR, Tugg Y, Marzok A, Miller MS. Beyond neutralization: Fc-dependent antibody effector functions in SARS-CoV-2 infection. *Nat Rev Immunol* 2023; **23**(6): 381-96.
9. Pierre CN, Adams LE, Anasti K, et al. Non-neutralizing SARS-CoV-2 N-terminal domain antibodies protect mice against severe disease using Fc-mediated effector functions. *bioRxiv* 2023.
10. Hariton E, Locascio JJ. Randomised controlled trials - the gold standard for effectiveness research: Study design: randomised controlled trials. *BJOG* 2018; **125**(13): 1716.
11. Gilbert PB, Fong Y, Hejazi NS, et al. Four statistical frameworks for assessing an immune correlate of protection (surrogate endpoint) from a randomized, controlled, vaccine efficacy trial. *Vaccine* 2024; **42**(9): 2181-90.
12. Bergeri I, Whelan MG, Ware H, et al. Global SARS-CoV-2 seroprevalence from January 2020 to April 2022: A systematic review and meta-analysis of standardized population-based studies. *PLoS Med* 2022; **19**(11): e1004107.
13. Gilbert PB, Donis RO, Koup RA, Fong Y, Plotkin SA, Follmann D. A Covid-19 Milestone Attained — A Correlate of Protection for Vaccines. *New England Journal of Medicine* 2022; **387**: 2203-6.
14. Hejazi NS, Shen X, Carpp LN, et al. Stochastic interventional approach to assessing immune correlates of protection: Application to the COVE messenger RNA-1273 vaccine trial. *Int J Infect Dis* 2023; **137**: 28-39.
15. Huang Y, Hejazi NS, Blette B, et al. Stochastic Interventional Vaccine Efficacy and Principal Surrogate Analyses of Antibody Markers as Correlates of Protection against Symptomatic COVID-19 in the COVE mRNA-1273 Trial. *Viruses* 2023; **15**(10): 2029.
16. Fong Y, McDermott AB, Benkeser D, et al. Immune Correlates Analysis of the ENSEMBLE Single Ad26.COVS Dose Vaccine Efficacy Clinical Trial. *Nature Microbiology* 2022; **7**(12): 1996-2010.
17. Hejazi NS, van der Laan MJ, Janes HE, Gilbert PB, Benkeser DC. Efficient nonparametric inference on the effects of stochastic interventions under two-phase sampling, with applications to vaccine efficacy trials. *Biometrics* 2021; **77**(4): 1241-53.

REVIEWERS' COMMENTS

Reviewer #1 (Remarks to the Author):

Thank you for your thoughtful revisions. The clarity of the manuscript has significantly improved with the added details on methods and discussions as suggested.

Reviewer #1 (Remarks on code availability):

The current README file does not cover every code file, making it difficult to follow the code flow. More comprehensive explanations would be helpful for readers.

Reviewer #2 (Remarks to the Author):

my comments have been addressed

REVIEWERS' COMMENTS

Reviewer #1 (Remarks to the Author):

Thank you for your thoughtful revisions. The clarity of the manuscript has significantly improved with the added details on methods and discussions as suggested.

Response: Thank you for the positive comments.

Reviewer #1 (Remarks on code availability):

The current README file does not cover every code file, making it difficult to follow the code flow. More comprehensive explanations would be helpful for readers.

Response: We agree that the manuscript would benefit from more comprehensive explanations of the code and thank you for bringing this to our attention. We have revised the Code Availability statement to provide more information on which code was used to generate the various tables and figures in the manuscript, as well as added references to the methods papers that describe the statistical methodologies. We have also added a README file to the Supplementary Software 2 file, which contains the code for conducting the stochastic interventional vaccine efficacy analysis. The revised Code Availability statement is provided below:

Code Availability

Code for generating Figure 1 and Supplementary Figure 3 is provided in the Supplementary Software 1 file.

Immune correlates analyses were done reproducibly based on publicly available R scripts.¹ The https://github.com/CoVPN/correlates_reporting2 repository hosts multiple modular workflows for correlates of risk/protection analyses and automated reporting of analytic results. Analysis modules used in the present work, along with the figures/tables they were used to generate, include: `cor_graphical` (graphical descriptions of correlates of risk: Figure 2 and Supplementary Figures 4-7); `cor_tabular` (tabular descriptions of correlates of risk: Table 1 and Supplementary Tables 2-5); `cor_coxph` (Cox proportional hazards modeling of risk: Table 2, Table 3, Figure 3, Supplementary Tables 11-18, Supplementary Figures 9-16, Supplementary Figures 41-48, Supplementary Figure 58); `immuno_graphical` (graphical descriptions of immunogenicity data: Supplementary Figure 8); `immuno_tabular` (tabular descriptions of immunogenicity data: Supplementary Tables 8, 9); and `cor_threshold` (nonparametric marker-thresholded correlate of risk modeling: Supplementary Figures 25-32). A complementary repository that handles the upstream data processing aspects of the analysis workflow is available at https://github.com/CoVPN/correlates_processing.

Code and documentation for nonparametric and Cox-based inference for controlled risk and controlled vaccine efficacy curves (Figure 4, Supplementary Figures 17-24, Supplementary Figures 33-40) and for calculating mediation effects for our application with no variability in the post vaccination antibody markers across placebo recipients (Supplementary Tables 19-22) is provided in the `vaccine` R package (version 1.2.1, <https://CRAN.R-project.org/package=vaccine>), with scripting code available at <https://github.com/Avi-Kenny/VaxCurve>. See also ref.²

Code for conducting the exposure-proximal analysis (Figure 5, Supplementary Figures 51-56) is publicly available at <https://github.com/yinghuang124/Exposure-Proximal>. See also ref.³

Code for conducting the stochastic interventional vaccine efficacy analysis (Supplementary Figures 49 and 50) is available in the Supplementary Software 2 file. This code draws heavily on the txshift R package, see <https://cran.r-project.org/package=txshift> and <https://github.com/nhejazi/txshift> for more extensive documentation. See also refs.⁴⁻⁶

1. Fong Y, Lu Y, Yu C, et al. CoVPN/correlates_reporting2: Neutralizing Antibody Correlate of Protection Against Severe-Critical COVID-19 in the ENSEMBLE Single-Dose Ad26.COV2.S Vaccine Efficacy Trial. Zenodo. DOI: 10.5281/zenodo.13690802. 2024.
2. Gilbert PB, Fong Y, Kenny A, Carone M. A Controlled Effects Approach to Assessing Immune Correlates of Protection. *Biostatistics* 2023;24:850–65.
3. Huang Y, Follmann D. Exposure proximal immune correlates analysis. kxae031, <https://doi.org/10.1093/biostatistics/kxae031>. *Biostatistics* 2024.
4. Hejazi NS, Shen X, Carpp LN, et al. Stochastic interventional approach to assessing immune correlates of protection: Application to the COVE messenger RNA-1273 vaccine trial. *Int J Infect Dis* 2023;137:28-39.
5. Hejazi NS, van der Laan MJ, Janes HE, Gilbert PB, Benkeser DC. Efficient nonparametric inference on the effects of stochastic interventions under two-phase sampling, with applications to vaccine efficacy trials. *Biometrics* 2021;77:1241-53.
6. Hejazi NS, Benkeser D. txshift: Efficient estimation of the causal effects of stochastic interventions in R. *Journal of Open Source Software* 2020;5:2447.

Reviewer #2 (Remarks to the Author):

my comments have been addressed

Response: Thank you, we feel our manuscript improved in the process of addressing them.